# SMARCA4/2 loss inhibits chemotherapy-induced apoptosis by restricting IP3R3-mediated Ca$^{2+}$ flux to mitochondria

Yibo Xue [1,2,3,4,18], Jordan L. Morris [5,18], Kangning Yang [1,2,18], Zheng Fu[1,2], Xianbing Zhu[1,2], Fraser Johnson[6,7,8], Brian Meehan[9], Leora Witkowski [3,10], Amber Yasmeen[11], Tunde Golenar[1,2], Mackenzie Coatham[12], Geneviève Morin[1,2], Anie Monast[1,2], Virginie Pilon[1,2], Pierre Olivier Fiset[13], Sungmi Jung[13], Anne V. Gonzalez[14], Sophie Camilleri-Broet [13], Lili Fu[13], Lynne-Marie Postovit[12,15], Jonathan Spicer [16], Walter H. Gotlieb[11], Marie-Christine Guiot[17], Janusz Rak [9], Morag Park [1,2], William Lockwood [6,7,8], William D. Foulkes [3,4,10], Julien Prudent [5,19✉] & Sidong Huang [1,2,19✉]

Inactivating mutations in *SMARCA4* and concurrent epigenetic silencing of *SMARCA2* characterize subsets of ovarian and lung cancers. Concomitant loss of these key subunits of SWI/SNF chromatin remodeling complexes in both cancers is associated with chemotherapy resistance and poor prognosis. Here, we discover that SMARCA4/2 loss inhibits chemotherapy-induced apoptosis through disrupting intracellular organelle calcium ion (Ca$^{2+}$) release in these cancers. By restricting chromatin accessibility to *ITPR3*, encoding Ca$^{2+}$ channel IP3R3, SMARCA4/2 deficiency causes reduced IP3R3 expression leading to impaired Ca$^{2+}$ transfer from the endoplasmic reticulum to mitochondria required for apoptosis induction. Reactivation of SMARCA2 by a histone deacetylase inhibitor rescues IP3R3 expression and enhances cisplatin response in SMARCA4/2-deficient cancer cells both in vitro and in vivo. Our findings elucidate the contribution of SMARCA4/2 to Ca$^{2+}$-dependent apoptosis induction, which may be exploited to enhance chemotherapy response in SMARCA4/2-deficient cancers.

[1] Department of Biochemistry, McGill University, Montreal, QC, Canada. [2] Rosalind and Morris Goodman Cancer Research Centre, McGill University, Montreal, QC, Canada. [3] Department of Human Genetics, McGill University, Montreal, QC, Canada. [4] Division of Medical Genetics, McGill University Health Centre, and Cancer Research Program, Research Institute of the McGill University Health Centre, McGill University, Montreal, QC, Canada. [5] Medical Research Council Mitochondrial Biology Unit, University of Cambridge, Cambridge, UK. [6] Department of Integrative Oncology, British Columbia Cancer Agency, Vancouver, BC, Canada. [7] Interdisciplinary Oncology Program, University of British Columbia, Vancouver, BC, Canada. [8] Department of Pathology and Laboratory Medicine, University of British Columbia, Vancouver, BC, Canada. [9] Department of Pediatrics, Research Institute of the McGill University Health Centre, Montreal Children's Hospital, McGill University, Montreal, QC, Canada. [10] Department of Specialized Medicine, Lady Davis Institute, Jewish General Hospital, McGill University, Montreal, QC, Canada. [11] Division of Gynecologic Oncology, Segal Cancer Center, Jewish General Hospital, McGill University, Montreal, QC, Canada. [12] Department of Oncology, Department of Obstetrics and Gynecology, University of Alberta, Edmonton, AB, Canada. [13] Department of Pathology, McGill University Health Centre, Montreal, QC, Canada. [14] Department of Medicine, Division of Respiratory Medicine, McGill University Health Centre, Montreal Chest Institute, Montreal, QC, Canada. [15] Department of Biomedical and Molecular Sciences, Queen's University, Kingston, ON, Canada. [16] Department of Surgery, McGill University Health Center, Montreal, QC, Canada. [17] Department of Pathology, Montreal Neurological Hospital/Institute, McGill University Health Centre, Montreal, QC, Canada. [18] These authors contributed equally: Yibo Xue, Jordan L. Morris, Kangning Yang. [19] These authors jointly supervised this work: Julien Prudent, Sidong Huang. ✉email: julien.prudent@mrc-mbu.cam.ac.uk; sidong.huang@mcgill.ca

The SWI/SNF family of ATP-dependent chromatin remodeling complexes control gene expression by regulating chromatin organization[1,2]. They also directly participate in DNA replication, repair, and recombination through modifying chromatin or recruiting relevant proteins[3]. Cancer genome-sequencing efforts have revealed mutations in SWI/SNF subunits in more than 20% of all human cancers, highlighting their critical roles in tumorigenesis[4]. However, identifying the driver mechanisms of SWI/SNF loss in promoting cancer remains a challenge.

SMARCA4 (BRG1) and SMARCA2 (BRM) are the two mutually exclusive ATPase subunits of SWI/SNF. SMARCA4 is inactivated by mutations or other mechanisms in ~10% of non-small cell lung cancer (NSCLC)[5–9]. Furthermore, concomitant loss of SMARCA4/2 protein expression occurs in a subset of NSCLC associated with a poor prognosis[6,10]. In addition to NSCLC, deleterious SMARCA4 mutations have been found to be the sole genetic driver in ~100% of small cell carcinoma of the ovary, hypercalcemic type (SCCOHT), a rare and aggressive ovarian cancer affecting young women[11–15]. SCCOHT is also characterized by concurrent loss of SMARCA4/2 protein expression, where SMARCA2 is epigenetically silenced and its reactivation strongly suppressed SCCOHT growth[16,17]. In contrast to other cancer types where experimental SMARCA2 inhibition is synthetic lethal with SMARCA4 loss[18–20], SMARCA2 silencing may cooperate with SMARCA4 loss in SMARCA4/2-deficient SCCOHT and NSCLC for cancer development[10,21]. However, the underlying mechanisms are not understood.

In addition to regulating gene expression, SWI/SNF components, including SMARCA4, have also been implicated in DNA-damage repair (DDR)[22–24]. Thus, their inactivation may also lead to compromised DDR and genome instability which are widely recognized as driving events in cancer development[25]. However, SCCOHT has a simple genome and harbors few mutations or chromosomal alterations other than inactivating mutations in SMARCA4[15,26,27], suggesting that altered transcriptional regulation may be the predominant driver of tumorigenesis in this cancer[28].

Platinum-based chemotherapies, such as cisplatin, induce DNA damage leading to cancer cell apoptosis and have been widely used in clinical practice for treating lung and ovarian cancers[29,30]. The involvement of SWI/SNF in DDR supports the use of these genotoxic agents for treating cancers with SMARCA4/2 deficiency, which does not often co-occur with other druggable oncogenic mutations. Indeed, previous studies have shown that experimental inhibition of SMARCA4 in SMARCA4-proficient cancer cells enhanced response to DNA damaging agents[31–33]. However, conventional chemotherapies are rarely effective for SCCOHT patients[15,34] and compared to other ovarian cancer types, SCCOHT cell lines show substantial resistance to these drugs[26,35]. In line with this, NSCLC patients with concomitant loss of SMARCA4/2 have a poorer prognosis than others[6,10] while adjuvant chemotherapy remains among primary treatment options for this cancer[29]. Thus, while SWI/SNF deficiencies have been widely associated to cancer progression, the mechanism by which SMARCA4/2-deficient cancer cells have adapted to resist chemotherapy is unknown.

In this study, we sought to examine the role of SMARCA4/2 in modulating chemotherapy responses in SCCOHT and NSCLC where SMARCA4/2 deficiency is frequently observed. Our results reveal a mechanism linking SMARCA4/2 loss to chemoresistance by inhibiting apoptosis induction and suggest a potential therapeutic strategy for improving treatment for SMARCA4/2-deficient cancers.

## Results

### SMARCA4/2 loss confers resistance to chemo-induced apoptosis in cancer cells

SCCOHT harbors few mutations or chromosomal alterations other than inactivating mutations in SMARCA4 but is typically resistant to conventional chemotherapy in patients[15,34], suggesting a potential connection between SMARCA4 deficiency and chemotherapy resistance. Since SMARCA4 is also frequently inactivated in NSCLC, we investigated the association of SMARCA4 expression with chemotherapy response in this cancer type. We first analyzed the most comprehensive NSCLC microarray gene expression data set with clinical outcome from the Director's Challenge data set of lung adenocarcinoma (LUAD, the most common NSCLC subtype) of diverse tumor staging[36]. For our analysis, we chose SMARCA4 "Jetset probe" unbiasedly identified by Kaplan–Meier (KM) plotter[37,38], which is the optimal probe set for specificity, coverage, and degradation resistance without preassociation with patient outcome. We stratified the patients within each treatment group based on median of SMARCA4 expression and found that low SMARCA4 expression was significantly associated with worse survival with adjuvant therapies (chemotherapy and radiation) when compared to high SMARCA4 expression (Supplementary Fig. 1a). This was supported by similar results obtained from KM plotter analyzing multiple available LUAD data sets of diverse tumor staging using the same probe (Supplementary Fig. 1b). A similar trend in UT lung SPORE data set[39] was also observed although not statistically significant (Supplementary Fig. 1c). Together, these patient outcome results suggest that SMARCA4 deficiency is associated with chemotherapy resistance in NSCLC, similar to that seen in SCCOHT.

Because patient outcomes from the data sets described above may be influenced by other variable factors such as treatment history, we next examined the role of SWI/SNF loss in mediating chemoresistance in more controlled experimental settings using cancer cell lines. First, we investigated the correlation between chemotherapy responses and mRNA expression levels of SMARCA4/2 in a large cohort of cell lines ($n = 436$) across different cancer types (Supplementary Fig. 2a), by integrating publicly available drug sensitivity data from Genomics of Drug Sensitivity in Cancer (GDSC)[40] and RNA sequencing (RNA-seq) data from Cancer Cell Line Encyclopedia (CCLE)[41,42]. We stratified these pan cancer cell lines ($n = 436$) based on their SMARCA4/2 expression in tertiles (Supplementary Fig. 2b) and found that SMARCA4$^{\text{Low}}$/SMARCA2$^{\text{Low}}$ ($A4^L/A2^L$, bottom tertile for both genes) group ($n = 53$) has the highest half maximal inhibitory concentration (IC$_{50}$) among all four groups, for common chemotherapy drugs with different mechanisms of action, including cisplatin, cyclophosphamide, topotecan, paclitaxel, etoposide, and 5FU (Fig. 1a, Supplementary Fig. 2c). Notably, IC$_{50}$ difference between $A4^L/A2^L$ and the SMARCA4$^{\text{High}}$/SMARCA2$^{\text{High}}$ ($A4^H/A2^H$ top tertile for both genes) group ($n = 50$) was statistically significant for all of these drugs. The SMARCA4$^{\text{Low}}$/SMARCA2$^{\text{High}}$ ($A4^L/A2^H$) group ($n = 24$) had the second highest IC$_{50}$ which was significantly higher than that of the $A4^H/A2^H$ group in three of the six drugs including cisplatin. We also observed a consistent trend of higher IC$_{50}$ in the SMARCA4$^{\text{High}}$/SMARCA2$^{\text{Low}}$ ($A4^H/A2^L$) group ($n = 34$) compared to $A4^H/A2^H$ although it was not statistically significant. Similar results were also obtained when analyzing lung cancer cell lines only (Fig. 1b, Supplementary Fig. 2d), which represented the largest cancer type ($n = 103$) among the CCLE panel (Supplementary Fig. 2a). Together, these observations show that reduced SMARCA4/2 expression correlates with resistance to different chemotherapies, including cisplatin, and suggest that SMARCA4 may play a dominant role in regulating drug responses in cancer cells.

To help unbiasedly assess the potential roles of SWI/SNF genes in modulating cisplatin responses, we performed a pooled CRISPR knockout screen targeting 496 epigenetic modifiers in OVCAR4, a SMARCA4/2-proficient high-grade serous ovarian

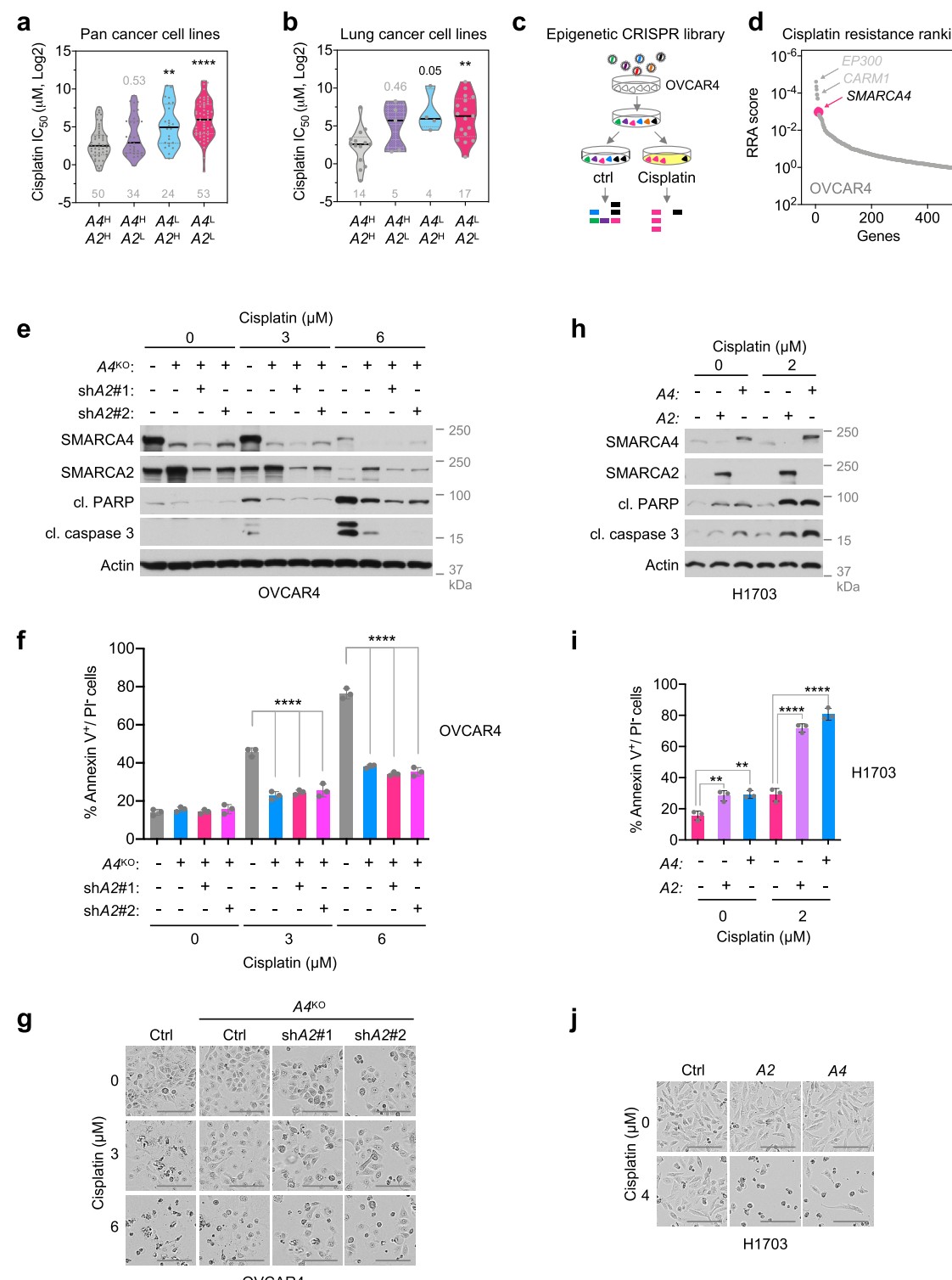

carcinoma (HGSC) cell line (Fig. 1c). Upon screen completion, we analyzed the data using the MAGeCK statistical software package[43,44] to search for candidate genes whose knockout may confer cisplatin resistance. Validating the screen, we identified *EP300* and *CARM1* among the top candidates (ranked #1 and #5, respectively; Supplementary Table 1 and Fig. 1d), whose suppression is known to confer cisplatin resistance[45–47]. In keeping with our above findings in patient outcome and CCLE cell line responses to chemotherapies, *SMARCA4* was also highly ranked (#11) in our screen suggesting that SMARCA4 loss

confers cisplatin resistance (Fig. 1c, d and Supplementary Table 1). *SMARCA2* was not significantly enriched (ranked #162), suggesting that SMARCA4 plays a dominant role in controlling cisplatin response, with SMARCA2 only compensating when SMARCA4 is lost.

To validate the above screen results, we knocked out *SMARCA4* in OVCAR4 cells using CRISPR/Cas9 genome editing system and investigated their apoptotic responses known to be induced by cisplatin treatment. Compared to the parental control, *SMARCA4* knockout ($A4^{KO}$) cells were more resistant to cisplatin-induced

**Fig. 1 SMARCA4/2 loss causes resistance to chemotherapeutics in ovarian and lung cancers.** The half maximal inhibitory concentration ($IC_{50}$) of cisplatin in pan cancer (**a**) and lung cancer (**b**) cell lines with differential mRNA expression for *SMARCA4* and *SMARCA2* (see Supplementary Fig. 2b for stratification). $A4^H$: *SMARCA4*High; $A4^L$: *SMARCA4*Low; $A2^H$: *SMARCA2*High; $A2^L$: *SMARCA2*Low. Cell line numbers are indicated in gray below each group. One-way ANOVA Kruskal–Wallis test followed by Dunn's test for multiple comparisons to $A4^HA2^H$ group, *p* values (*p*): **a** $A4^HA2^L$—0.5338, $A4^LA2^H$—0.0035, $A4^LA2^L < 0.0001$; **b** $A4^HA2^L$—0.4615, $A4^LA2^H$—0.0517, $A4^LA2^L$—0.0019. **c** Schematic outline of a pooled CRISPR screen with a sgRNA knockout library against epigenetic regulators to identify genes required for cisplatin response in OVCAR4 cells. **d** MAGeCK analysis[43,44,81] for screen in **c**. Genes were ranked by robust rank aggregation (RRA). Immunoblots (**e**), annexin $V^+/PI^-$ apoptotic cell population determined by flow cytometry (**f**), and representative phase-contrast images (**g**) of OVCAR4 cells with indicated *SMARCA4/2* perturbations and cisplatin treatments (**e**, **f** 48 h). Immunoblots (**h**), annexin $V^+/PI^-$ apoptotic cell population (**i**), and representative phase-contrast images (**j**) of H1703 cells with indicated *SMARCA4/2* perturbations and cisplatin treatments (**h**, **i** 72 h). **e–j** Ctrl Control, $A4^{KO}$ *SMARCA4* knockout, sh*A2* shRNA targeting *SMARCA2*, cl. PARP cleaved PARP, cl. caspase 3 cleaved caspase 3, A4 SMARCA4, A2 SMARCA2. Scale bar, 150 μm. Mean ± SD, $n = 3$ independent experiments, one-way ANOVA followed by Dunnett's test for multiple comparisons, *p* values (*p*): **f** all ($<0.0001$); **i** A2 (0 μM)—0.0032, A4 (0 μM)—0.0023, A2 or A4 (3 μM) $< 0.0001$. ***p < 0.01**, ****p < 0.0001$.

elevation of annexin V (cell death marker; Supplementary Fig. 3a), cleaved PARP, and cleaved caspase 3 (apoptosis markers; Fig. 1e). They also exhibited reduced annexin $V^+$/propidium iodide $(PI)^-$ apoptotic cell population (Fig. 1f), and had fewer morphological defects, a characteristic of the apoptotic cell (Fig. 1g) in response to cisplatin treatment. Similarly, *SMARCA4* knockout also protected OVCAR4 cells against paclitaxel-induced apoptosis (Supplementary Fig. 3b). Furthermore, knockdown of *SMARCA2* using two independent short hairpin RNAs (shRNAs) in these $A4^{KO}$ cells led to increased resistance to the above-described apoptotic responses induced by cisplatin (Fig. 1e–g, Supplementary Fig. 3a). Similar results were obtained in the HEC116 ovarian endometrial cancer cell line (Supplementary Fig. 3c, d), further validating the above results in OVCAR4 cells. We also noted that high dose of cisplatin treatment in OVCAR4 control cells led to reduced SMARCA4/2 protein expression (Fig. 1e), suggesting a potential negative feedback regulation or a selection for cells expressing low SMARCA4/2. To corroborate our results, we sought to perform the reverse experiments by restoring SMARCA4 or SMARCA2 in SMARCA4/2-deficient cancer cells. SMARCA4/2 restoration in SCCOHT cells both strongly suppressed their growth[16,17], which limited the experimental window to study apoptosis regulation upon subsequent cisplatin treatment. In contrast, SMARCA4/2-deficient NSCLC cells including H1703 cells can tolerate restoration of SMARCA4/2[48] and thus are better suited for this analysis. Ectopic expression of SMARCA4 or SMARCA2 sensitized H1703 cells to cisplatin treatment and led to strong induction of apoptosis, indicated by elevation of annexin V, cleaved PARP, and cleaved caspase 3, a marked increase of the annexin $V^+/PI^-$ apoptotic cell population, acquisition of apoptotic cell morphology, and impaired growth (Fig. 1h–j, Supplementary Fig. 3e, f). Further supporting this, CRISPR/Cas9-mediated *SMARCA4* knockout in SMARCA4/2-proficient H1437 NSCLC cancer cells conferred resistance to apoptosis induced by cisplatin treatment; knockdown of *SMARCA2* in these $A4^{KO}$ cells led to further increased resistance to cisplatin, indicated by reduction of cleaved PARP and cleaved caspase 3 and increased cell viability (Supplementary Fig. 3g, h).

We further examined the effect of SMARCA4 loss in response to other common chemotherapeutics using above-described isogenic cell pairs of HEC116 and H1703 that differ only in SMARCA4 status. Consistent with cisplatin results, *SMARCA4* knockout in HEC116 cells suppressed elevation of cleaved PARP and cleaved caspase 3 induced by cyclophosphamide, topotecan, and paclitaxel (Supplementary Fig. 4a) and led to increased cell viability in the presence of these agents (Supplementary Fig. 4b). Conversely, SMARCA4 restoration sensitized H1703 cells to the treatment with these drugs, as indicated by elevation of apoptosis and increased cell viability (Supplementary Fig. 4c, d). Together, our data indicate that SMARCA4/2 loss inhibits chemotherapy-induced apoptotic responses in ovarian and lung cancer cells.

**SMARCA4/2 loss results in altered intracellular $Ca^{2+}$ homeostasis in cancer cells**. To understand how SMARCA4/2 regulate chemotherapy sensitivity and apoptosis induction, we analyzed the transcriptome regulated by SMARCA4 using SCCOHT cells, taking advantage of their simple genetic background. Gene set enrichment analysis (GSEA) of RNA-seq data generated in SCCOHT-1 and BIN-67 cells ± SMARCA4 restoration[49] reveals top ten Gene Ontology (GO) terms regulated by SMARCA4 consistently shared by these two SCCOHT cell lines (Supplementary Fig. 5). Multiple terms associated with ion/calcium homeostasis were identified including "ion transmembrane transporter" and "calcium ion binding" (Fig. 2a, b). The established crucial role of calcium ion ($Ca^{2+}$) homeostasis in apoptosis induction[50] makes these GO terms particularly interesting. Transient $Ca^{2+}$ release from the endoplasmic reticulum (ER), the major intracellular $Ca^{2+}$ store, into the cytosol and subsequent transfer to mitochondria is important for cellular signal transductions as well as ATP production[51]. However, excessive ER-$Ca^{2+}$ release leads to mitochondrial $Ca^{2+}$ overload and cell death, which has recently been associated to the selective vulnerability of cancer cells[52–54]. Together, these transcriptome analyses in SCCOHT cell lines indicate that $Ca^{2+}$ homeostasis may be a commonly altered cellular process by SMARCA4, contributing to their roles in apoptosis regulation and cancer cell survival.

Given the crucial role of intracellular $Ca^{2+}$ signaling in apoptosis induction, we reasoned that SMARCA4/2 may affect apoptosis by regulating intracellular $Ca^{2+}$ flux. To validate the role of SMARCA4/2 in $Ca^{2+}$ homeostasis and transfer to mitochondria, we measured the changes in cytosolic and mitochondrial $Ca^{2+}$ content of SCCOHT-1 cells, ±SMARCA4 restoration, in response to histamine, an inositol trisphosphate (IP3) agonist activating ER-$Ca^{2+}$ release via inositol trisphosphate receptor (IP3R)[55]. In order to monitor intracellular $Ca^{2+}$ dynamics, we expressed genetically encoded $Ca^{2+}$ indicators (GECI) targeted to the cytosol (R-GECO)[56] or mitochondria (CEPIA-2mt)[57] and monitored GECI fluorescence upon ER-$Ca^{2+}$ release stimulation by spinning disk confocal microscopy (Supplementary Fig. 6). While histamine stimulation induced little changes in cytosolic or mitochondrial $Ca^{2+}$ in SCCOHT-1 control cells, it strongly elevated $Ca^{2+}$ content in both compartments in SMARCA4-restored cells (Fig. 2c–e). Consistent with this, restoration of SMARCA4 in H1703 cells also significantly increased ER-$Ca^{2+}$ release to the cytosol and $Ca^{2+}$ transfer to the mitochondria upon histamine stimulation, compared to control cells (Fig. 2f–h). These data indicate that SMARCA4 plays a causal role in regulating intracellular $Ca^{2+}$ homeostasis by enabling ER-$Ca^{2+}$ release to the cytosol and mitochondria.

The increased cytosolic and mitochondrial $Ca^{2+}$ content observed upon SMARCA4 restoration could be due to either direct enhanced $Ca^{2+}$ release from the ER or elevated capacity of the ER-$Ca^{2+}$ content. To distinguish these possibilities, we measured the cytosolic $Ca^{2+}$ changes in above isogenic cell pairs of SCCOHT-1

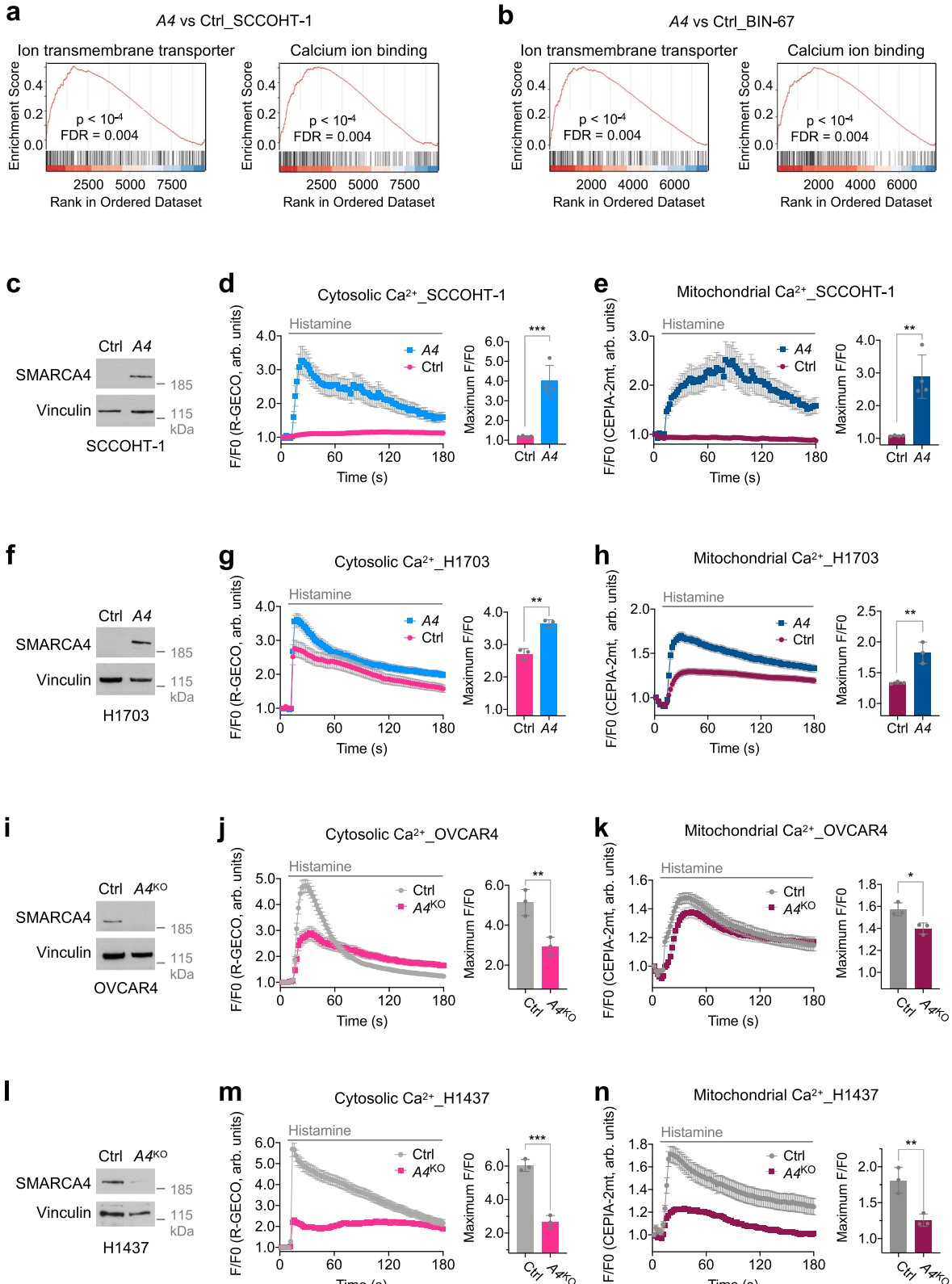

and H1703 in response to thapsigargin, an inhibitor of sarcoplasmic/ER Ca²⁺-ATPase, which can entirely deplete ER-Ca²⁺ stores[55]. Interestingly, restoration of SMARCA4 does not increase maximal cytosolic ER-Ca²⁺ release induced by thapsigargin treatment in SCCOHT-1 or H1703 cells (Supplementary Fig. 7a, b), suggesting that SMARCA4 promotes Ca²⁺ release from the ER rather than an increase in ER-Ca²⁺ storage capacity. Further supporting this,

SMARCA4 knockout in OVCAR4 and H1437 cells significantly decreased the induction of cytosolic and mitochondrial Ca²⁺ upon histamine treatment (Fig. 2i–n), even though *SMARCA4* knockout had increased ER-Ca²⁺ stores as indicated by an increase in cytosolic Ca²⁺ in OVCAR4 cells, but not in H1437 cells, following thapsigargin stimulation (Supplementary Fig. 7c, d). Finally, to rule out the potential contribution of the mitochondrial Ca²⁺ uptake

**Fig. 2 SMARCA4 modulates Ca$^{2+}$ flux from the ER to mitochondria.** Gene set enrichment analysis plots of indicated Gene Ontology terms in SCCOHT-1 (**a**) and BIN-67 (**b**) cells ± *SMARCA4* (*A4*) restoration[49]. Ctrl control, FDR false discovery rate. **c** Immunoblots of indicated proteins in SCCOHT-1 cells ± *A4* restoration. Changes of cytosolic (**d**) and mitochondrial (**e**) Ca$^{2+}$ contents in SCCOHT-1 cells ± *A4* restoration upon histamine stimulation. **d** 28 Ctrl and 21 *A4* cells from $n = 4$ independent experiments were analyzed. **e** 44 Ctrl and 20 *A4* cells from $n = 4$ independent experiments were analyzed. **f** Immunoblots of indicated proteins in H1703 cells ± *A4* restoration. Changes of cytosolic (**g**) and mitochondrial (**h**) Ca$^{2+}$ contents in H1703 cells ± *A4* restoration upon histamine stimulation. **g** 41 Ctrl and 74 *A4* cells from $n = 3$ independent experiments were analyzed. **h** 45 Ctrl and 63 *A4* cells from $n = 3$ independent experiments were analyzed. **i** Immunoblots of indicated proteins in OVCAR4 cells ± SMARCA4 knockout (*A4*$^{KO}$). Changes of cytosolic (**j**) and mitochondrial (**k**) Ca$^{2+}$ contents in OVCAR4 cells with ± *A4*$^{KO}$ upon histamine stimulation. **j** 60 Ctrl and 53 *A4*$^{KO}$ cells from $n = 3$ independent experiments were analyzed. **k** 41 Ctrl and 40 *A4*$^{KO}$ cells from $n = 3$ independent experiments were analyzed. **l** Immunoblots of indicated proteins in H1437 cells ± *A4*$^{KO}$. Changes of cytosolic (**m**) and mitochondrial (**n**) Ca$^{2+}$ contents in H1437 cells ± *A4*$^{KO}$ upon histamine stimulation. **m** 39 Ctrl and 37 *A4*$^{KO}$ cells from $n = 3$ independent experiments were analyzed. **n** 38 Ctrl and 42 *A4*$^{KO}$ cells from $n = 3$ independent experiments were analyzed. **d**, **e**, **g**, **h**, **j**, **k**, **m**, **n** Left: traces of cytosolic and mitochondrial Ca$^{2+}$ contents upon 100 μM histamine stimulation (mean ± SEM). Right: quantification of the maximal Ca$^{2+}$ signal peaks induced by histamine stimulation (mean ± SD). The Ca$^{2+}$ probes R-GECO (R-GECO F/F0) and CEPIA-2mt (CEPIA-2mt F/F0) were used to monitor cytosolic and mitochondrial Ca$^{2+}$, respectively. Arb. units arbitrary units. Two-tailed *t*-test, *p* values (*p*): **d** 0.0003, **e** 0.0016, **g** 0.0012, **h** 0.0084, **j** 0.0088, **k** 0.0182, **m** 0.0004, **n** 0.0084. **p* < 0.05, ***p* < 0.01, ****p* < 0.001.

machinery in this phenotype, we showed that protein levels of the mitochondrial calcium uniporter (MCU) and its regulators[58,59] were unchanged in these cell lines, indicating that Ca$^{2+}$ transfer defects were not due to defective mitochondrial Ca$^{2+}$ import machinery (Supplementary Fig. 7e). Together, these results suggest that SMARCA4/2 regulate intracellular Ca$^{2+}$ homeostasis and mitochondrial Ca$^{2+}$ content likely by controlling Ca$^{2+}$ release from the ER.

**SMARCA4/2 directly regulate *ITPR3* expression.** To dissect the detailed mechanism by which SMARCA4/2 regulate Ca$^{2+}$ homeostasis, we further investigated Ca$^{2+}$-related genes in ion/calcium associated GO terms identified from the above transcriptome analysis in SCCOHT cells (Fig. 2a, b). Overlapping the two datasets yielded 198 common genes affected by SMARCA4 restoration in both SCCOHT-1 and BIN-67 cells (Fig. 3a; Supplementary Table 2). To help identify direct targets of SMARCA4, we examined these 198 commonly regulated genes in a chromatin immunoprecipitation sequencing (ChIP-seq) data set profiling SMARCA4 occupancy in BIN-67 cells ± SMARCA4 restoration[60]. This analysis revealed 69 of the 198 genes showing SMARCA4 occupancy in their loci (Fig. 3a; Supplementary Table 2). Considering that SMARCA4 and SMARCA2 may regulate the same target genes and that SMARCA4 also modulates Ca$^{2+}$ homeostasis in NSCLC cells (Fig. 2f–h), we then examined the regulation of these 69 genes in an independent RNA-seq data set of BIN-67 cells ± SMARCA4/2 restoration[60] and a microarray data set of NSCLC cell line H1299 ± SMARCA4 restoration[61]. Notably, all of the 69 SMARCA4-affected genes were also regulated by SMARCA2 in BIN-67 cells, indicating that SMARCA4/2 may have redundant function in controlling Ca$^{2+}$ homeostasis (Fig. 3b). In keeping with the fact that lung cancer cells have more complex genetic landscapes than SCCOHT[15,62], only four genes, namely *ITPR3*, *MATN2*, *EHD4*, and *ATP2B4*, were consistently upregulated by SMARCA4 in both cancer types (Fig. 3b).

Among these four common genes, *ITPR3* encodes inositol 1,4,5-trisphosphate receptor type 3 (IP3R3), one of the IP3R family members that forms Ca$^{2+}$ channels on the ER and plays critical roles in intracellular Ca$^{2+}$ homeostasis and cell apoptosis[52,63]. IP3R3 localizes at the mitochondria-associated membranes, a signaling platform allowing the generation of microdomains of high Ca$^{2+}$ concentration required for efficient mitochondrial Ca$^{2+}$ uptake[64], and preferentially transmits apoptotic Ca$^{2+}$ signals into mitochondria over other IP3Rs[65]. Tumor suppressors such as PTEN, BAP1, and PML have been shown to induce apoptosis in cancer cell by promoting IP3R3-mediated Ca$^{2+}$ flux from the ER to mitochondria[66–68]. Thus, we hypothesized that SMARCA4/2 may promote Ca$^{2+}$ flux to the mitochondria and apoptosis

induction by directly regulating *ITPR3* gene expression. Corroborating our transcriptome data above (Fig. 3b), ectopic expression of SMARCA4 or SMARCA2 in both SCCOHT (BIN-67, SCCOHT-1) and NSCLC (H1299, H1703) cells resulted in elevated mRNA and protein expression of IP3R3 (Fig. 3c, d). Conversely, *SMARCA4* knockout in OVCAR4, HEC116, and H1437 cells suppressed IP3R3 expression which was further downregulated upon subsequent *SMARCA2* knockdown (Supplementary Fig. 8). These data established that SMARCA4/2 promote IP3R3 expression in both ovarian and lung cancer cells, likely through direct regulation of transcription.

Given the chromatin remodeling role of SWI/SNF, we then focused on the chromatin architecture of the *ITPR3* locus and its potential regulation by SMARCA4/2. Indeed, SMARCA4 occupancy was observed at the *ITPR3* promoter in ChIP-seq data of the BIN-67 cells upon SMARCA4 restoration (Fig. 3e)[60]. We also detected this SMARCA4 occupancy in H1703 cells with SMARCA4 restoration[48] and in H1299 cells expressing inducible SMARCA4[61] (Fig. 3e). These data suggest that SMARCA4/2 may directly regulate *ITPR3* expression. Consistent with this, we found that ChIP-seq signals of H3K27Ac, a chromatin mark associated with active promoter and enhancer, were elevated at the upstream and gene body regions of *ITPR3* in BIN-67 cells after SMARCA4 restoration[60] and in H1703 cells after restoration of SMARCA4 or SMARCA2[48] (Fig. 3f, upper panel). Furthermore, the assay for transposase-accessible chromatin using sequencing (ATAC-seq) peaks at these *ITPR3* genomic regions were also elevated upon SMARCA4/2 restoration in H1703 cells (Fig. 3f, lower panel), indicating an enhanced chromatin accessibility at the *ITPR3* locus when SMARCA4/2 were present. Together, these data suggest that SMARCA4/2 promote *ITPR3* transcription by directly remodeling chromatin structure at its gene locus.

**SMARCA4/2 loss inhibits apoptosis by restricting IP3R3-mediated Ca$^{2+}$ flux to mitochondria.** Next, we investigated whether reduced IP3R3 expression accounts for compromised Ca$^{2+}$ flux in SMARCA4/2-deficient SCCOHT and NSCLC cells (Fig. 2c–n). To this end, we performed rescue experiments by suppressing SMARCA4-mediated IP3R3 induction in SCCOHT-1 and H1703 cells. Accompanied by an increase of IP3R3 levels (Fig. 4a), ectopic SMARCA4 expression in SCCOHT-1 cells strongly elevated cytosolic (Fig. 4b) and mitochondrial (Fig. 4c) Ca$^{2+}$ contents in response to histamine stimulation. Notably, in these SMARCA4-restored cells, shRNA-mediated knockdown of IP3R3 to levels similar to control cells prevented ER-Ca$^{2+}$ release, characterized by a significant decrease of cytosolic and mitochondrial Ca$^{2+}$ contents (Fig. 4a–c). These results were confirmed in H1703 cells where suppression of IP3R3 was achieved

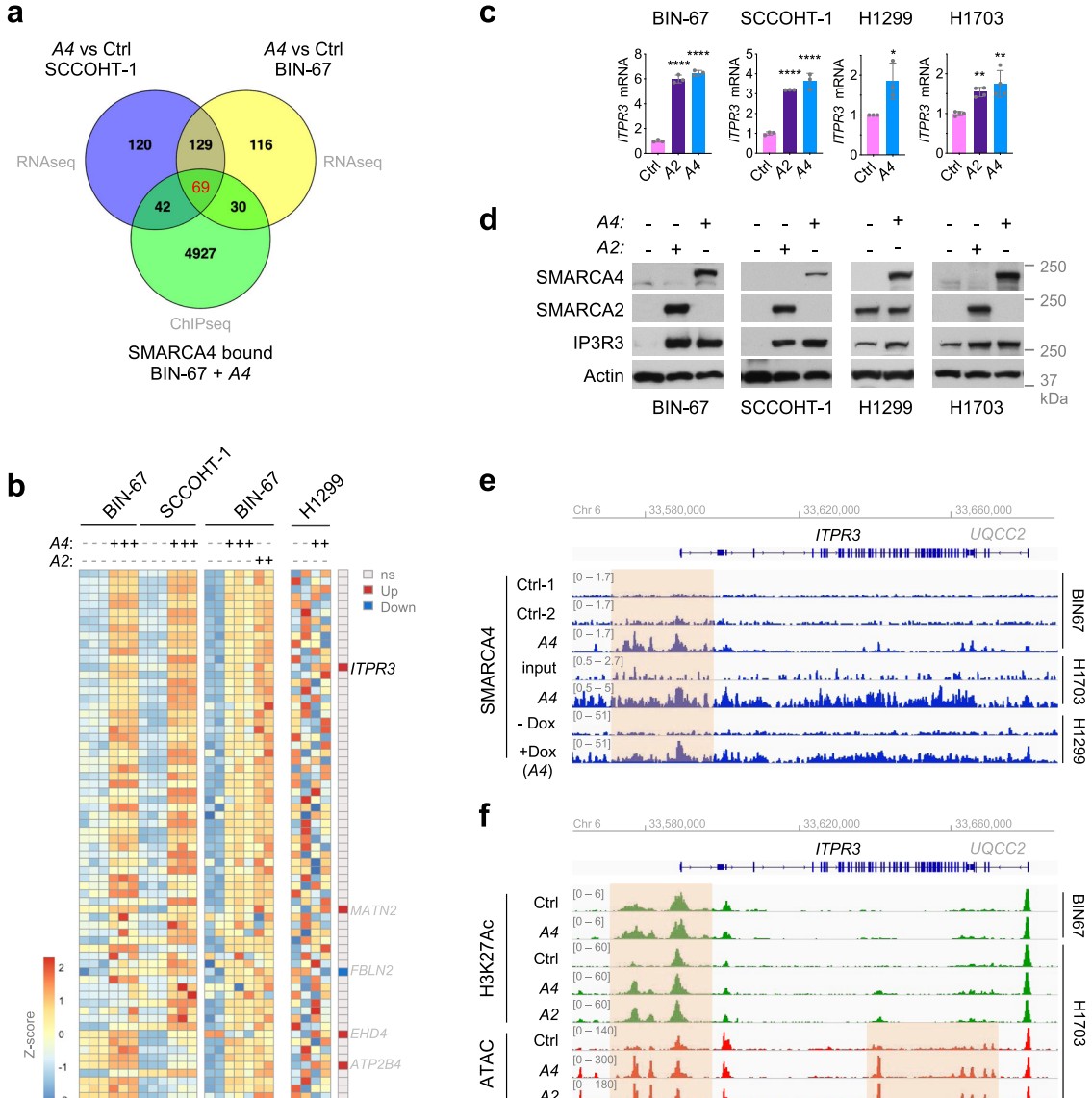

**Fig. 3 SMARCA4/2 regulate ITPR3 transcription through remodeling chromatin accessibility at its gene locus. a** Venn diagram of Ca$^{2+}$-related genes from Fig. 2a, b that are enriched in SCCOHT-1 and BIN-67 cells with SMARCA4 restoration. **b** Heatmap of Ca$^{2+}$-related genes bound by SMARCA4 ($n = 69$) in indicated SCCOHT (SCCOHT-1 and BIN-67) and NSCLC (H1299) cell lines with SMARCA4/2 restoration. Left: normalized reads from RNA-seq data of BIN-67 and SCCOHT-1 cells with SMARCA4 restoration[49]. Middle: normalized reads from RNA-seq data of BIN-67 cells with SMARCA4/2 restoration[60]. Right: normalized signal from microarray data of H1299 cells with SMARCA4 restoration[61]. Row scaling was used to generate the heatmap. The last column represents changes of genes in H1299 cells ± SMARCA4 restoration: ns not significant, up upregulated, down downregulated. **c** RT-qPCR measurements of *ITPR3* mRNA expression in indicated SCCOHT and NSCLC cell lines with SMARCA4/2 restoration. GAPDH was used for normalization. Mean ± SD, $n = 3$ (BIN-67, SCCOHT-1, H1299) or 4 (H1703) independent experiments, one-way ANOVA followed by Dunnett's test for multiple comparisons to the control group (BIN-67, SCCOHT-1, H1703) or two-tailed *t*-test (H1299), *p* values (*p*): BIN-67, all <0.0001; SCCOHT-1, A2—0.0004, A4—0.0001; H1299—0.0318; H1703, A2—0.0097, A4—0.0013. **d** Immunoblots of indicated proteins in indicated SCCOHT and NSCLC cell lines ± SMARCA4/2 restoration. **e** SMARCA4 occupancy in vicinity of the *ITPR3* locus assessed by chromatin immunoprecipitation sequencing (ChIP-seq) in indicated SCCOHT and lung cancer cell lines ± SMARCA4 restoration. SMARCA4 in H1299 cells was induced by doxycycline (Dox)[61]. Track height is normalized to relative number of mapped reads. **f** Chromatin structure changes in vicinity of the ITPR3 locus assessed by H3K27Ac ChIP-seq and assay for transposase-accessible chromatin sequencing (ATAC-seq) in indicated SCCOHT and lung cancer cell lines ± SMARCA4/2 restoration. Track height is normalized to relative number of mapped reads. **a–f** Ctrl control, *A4* SMARCA4, *A2* SMARCA2. *$p < 0.05$, **$p < 0.01$, ****$p < 0.0001$.

by small-interfering RNA (siRNA) (Fig. 4d–f). Furthermore, cytosolic Ca$^{2+}$ measurement upon thapsigargin stimulation in the above SCCOHT-1 and H1703 cells indicated that ER-Ca$^{2+}$ storage capacity was not significantly altered upon *ITPR3* knockdown (Supplementary Fig. 9). Together, these data indicate that reduced IP3R3 expression is the critical contributor to the compromised Ca$^{2+}$ flux in SMARCA4/2-deficient cells.

In line with the established role of IP3R3 in Ca$^{2+}$-mediated apoptosis, suppression of IP3R3 in OVCAR4 cells prevented cisplatin-induced apoptosis as indicated by reduced levels of cleaved PARP and cleaved caspase 3 (Fig. 4g) and the annexin V$^+$/PI$^-$ apoptotic cell population (Fig. 4h). Conversely, ectopic expression of IP3R3 in H1703 cells enhanced apoptotic induction and growth suppression after cisplatin treatment (Fig. 4i,

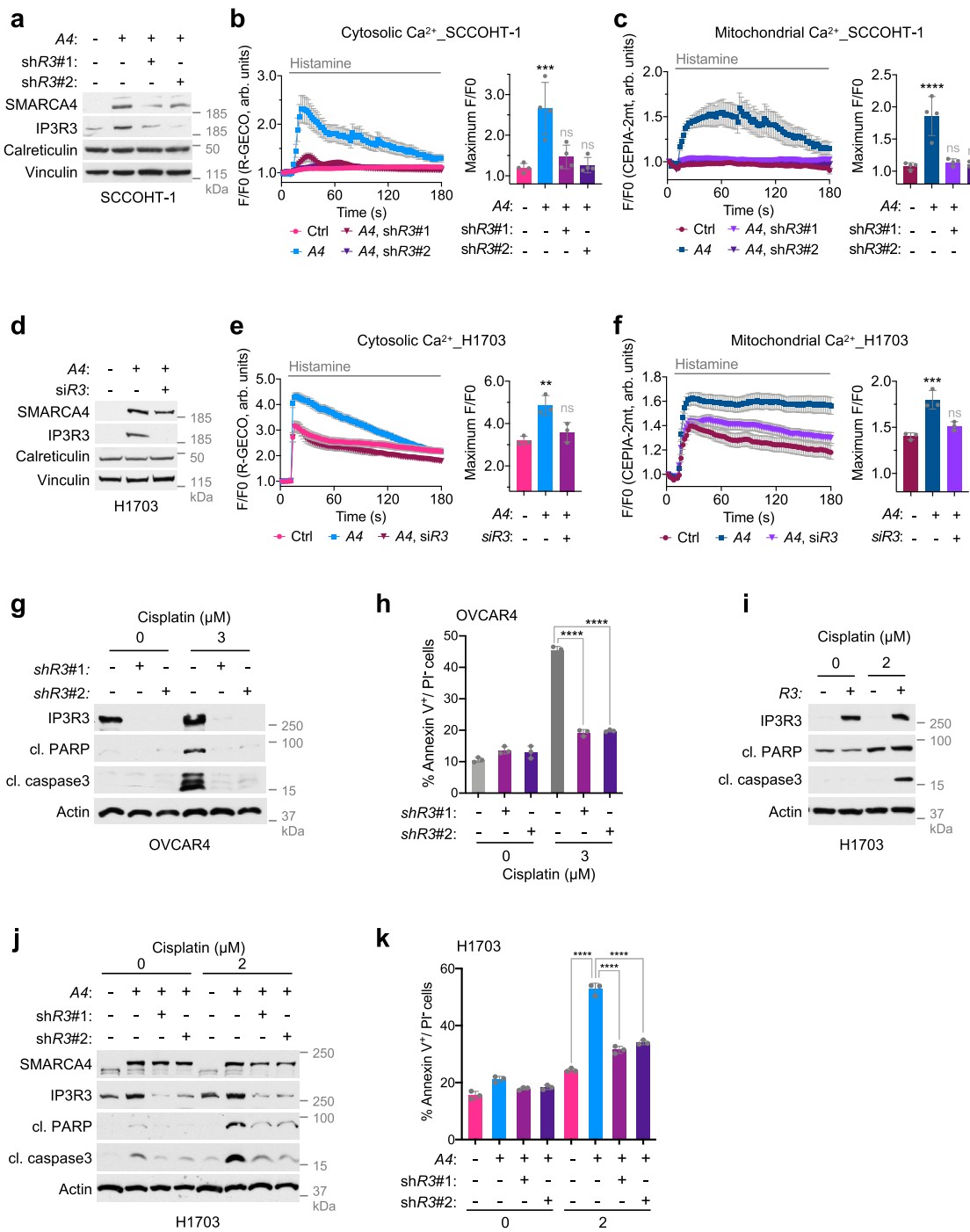

Supplementary Fig. 10a, b). Similarly, ectopic IP3R3 expression also sensitized BIN-67 cells to cisplatin treatment (Supplementary Fig. 10c, d). Thus, IP3R3 seems to be necessary and sufficient to mediate cisplatin-induced apoptosis in these models. Given that SMARCA4/2 directly activates *ITPR3* expression (Fig. 3), we then investigated whether reduced IP3R3 expression in SMARCA4/2-deficient cells drives resistance to chemotherapy-induced apoptosis. As shown in Fig. 4j, k, while SMARCA4 restoration in H1703 cells led to increased IP3R3 expression with concomitant elevation of cleaved PARP and cleaved caspase 3 as well as the annexin V[+]/PI[−] apoptotic cell population after cisplatin treatment, knockdown of IP3R3 markedly suppressed the induction of these apoptosis markers in these SMARCA4-expressing cells,

corroborating $Ca^{2+}$ signaling defects in these cells (Fig. 4d–f). Together, these data suggest that SMARCA4/2 loss inhibits chemotherapy-induced apoptosis by constricting IP3R3-mediated $Ca^{2+}$ flux to mitochondria.

**IP3R3 expression is reduced in SMARCA4/2-deficient cancers.** To further validate our findings of *ITPR3* regulation by SMARCA4/2 in cell models with genetic perturbation, we analyzed mRNA expression of *ITPR3* and *SMARCA4/2* in RNA-seq data sets of ovarian ($n = 47$) and lung cancer ($n = 192$) cell lines available from CCLE[41,42]. For both cancer types, cell lines with low *SMARCA4* expression (bottom quartile) also expressed lower

**Fig. 4 SMARCA4/2 loss inhibits apoptosis by constricting IP3R3-mediated Ca$^{2+}$ flux. a** Immunoblots of SCCOHT-1 cells ± *SMARCA4* (*A4*) and *ITPR3* (*R3*) perturbations. sh*R3*: shRNA targeting *ITPR3*. Changes of cytosolic (**b**) and mitochondrial (**c**) Ca$^{2+}$ contents in SCCOHT-1 cells ± *A4* and *R3* perturbations upon histamine stimulation. **b** 43 control (Ctrl), 30 *A4*, 51 *A4* Sh*R3*♯1, and 50 *A4* Sh*R3*♯2 cells from $n = 4$ independent experiments were analyzed. **c** 31 Ctrl, 30 *A4*, 50 *A4* Sh*R3*♯1, and 50 *A4* Sh*R3*♯2 cells from $n = 4$ independent experiments were analyzed. **d** Immunoblots of H1703 cells ± *A4* and *R3* perturbations. si*R3* siRNA targeting *ITPR3*. Changes of cytosolic (**e**) and mitochondrial (**f**) Ca$^{2+}$ contents in H1703 cells ± *A4* and *R3* perturbations upon histamine stimulation. **e** 64 Ctrl, 70 *A4*, and 64 *A4* si*R3* cells from $n = 3$ independent experiments were analyzed. **f** 53 Ctrl, 53 *A4*, and 50 *A4* si*R3* cells from $n = 3$ independent experiments were analyzed. **g** Immunoblots of OVCAR4 cells with *ITPR3* knockdown 48 h post cisplatin treatment. cl cleaved. **h** Annexin V$^+$/PI$^-$ apoptotic cell population determined by flow cytometry in OVCAR4 cells described in **g**. Immunoblots of H1703 cells ± *R3* overexpression (**i**) or ±*A4* and *R3* perturbations (**j**). Cells were collected 72 h after the treatment. **k** Annexin V$^+$/PI$^-$ apoptotic cell population determined by flow cytometer in H1703 cells described in **j**. **b**, **c**, **e**, **f** Left: traces of cytosolic or mitochondrial Ca$^{2+}$ contents upon 100 μM histamine stimulation (mean ± SEM). Right: quantification of the maximal Ca$^{2+}$ signal peaks induced by histamine (mean ± SD). The Ca$^{2+}$ probes R-GECO (R-GECO F/F0) and CEPIA-2mt (CEPIA-2mt F/F0) were used to monitor cytosolic and mitochondrial Ca$^{2+}$, respectively. Arb. units arbitrary units. One-way ANOVA followed by Dunnett's test for multiple comparisons to Ctrl, p values (p): **b** A4—0.0003, shR3#1—0.6223, shR3#2—0.9866; **c** A4 < 0.0001, shR3#1—0.9109, shR3#2—0.9845; **e** A4—0.0038, siR3—0.4232; **f** A4—0.0007, siR3—0.1620. **h, k** Mean ± SD, $n = 3$ independent experiments, one-way ANOVA followed by Dunnett's test for multiple comparisons, p values (p): all <0.0001. *p < 0.01, ***p < 0.001, ****p < 0.0001; ns not significant.

levels of *ITPR3* compared to the rest of cell lines with high *SMARCA4* expression (Supplementary Fig. 11a). Furthermore, we observed a significant positive correlation between *ITPR3* and *SMARCA2* in these ovarian ($n = 11$, $r = 0.825$) and lung ($n = 48$, $r = 0.584$) cancer cell lines with low *SMARCA4* expression (Fig. 5a). Moreover, in a panel of 20 NSCLC cell lines, reduced IP3R3 protein was observed in SMARCA4-deficient cells compared to SMARCA4-proficient cells; overall SMARCA4/2 dual deficient cell lines expressed the lowest levels of IP3R3 (Fig. 5b). These results are in line with our above functional data, supporting that IP3R3 expression is reduced in SMARCA4/2-deficient ovarian and lung cancer cells.

Next, we investigated the relationship between IP3R3 and SMARCA4/2 expression in patient tumors. We analyzed the available TCGA RNA-seq data sets of ovarian serous cystadenocarcinoma (OV)[69], LUAD, and lung squamous cell carcinoma tumors[7,70]. Similar to the above observations in cell lines, *ITPR3* mRNA in patient tumors with the bottom quartile of *SMARCA4* expression is significantly reduced compared the other tumors in all three data sets (Supplementary Fig. 11b). Confirming the cell line results (Fig. 5a), *ITPR3* was also significantly correlated with *SMARCA2* mRNA in these tumors with low *SMARCA4* expression (Fig. 5c, Supplementary Fig. 11c). Furthermore, we analyzed *ITPR3* mRNA expression in SCCOHT patient tumors ($n = 13$) characterized by concomitant loss of SMARCA4/2 protein expression. In keeping with above analysis, *ITPR3* mRNA in SCCOHT tumors is similar to OV tumors with low expression of *SMARCA4/2* ($n = 42$) while significantly lower than OV tumors with high expression of *SMARCA4/2* ($n = 50$)[69] (Fig. 5d). Using immunohistochemistry (IHC), we also examined IP3R3 protein expression in patient tumors of SCCOHT and HGSC with an IP3R3 antibody whose IHC specificity was verified by RNAi (Supplementary Fig. 12). As shown in Fig. 5e, f, SCCOHT tumors ($n = 45$) expressed significantly lower levels of IP3R3 than HGSCs ($n = 45$). Consistently, NSCLC tumors with low SMARCA4 expression ($n = 9$, $H$-score ≤ 100) expressed significantly lower IP3R3 protein than those with higher SMARCA4 expression ($n = 50$, $H$-score > 200) (Fig. 5g, h). Together, these results from multiple cohorts of cell lines and patient tumor samples support the cooperative roles of SMARCA4/2 in regulating *ITPR3* and confirm reduced IP3R3 expression in SMARCA4/2-deficient cancers.

Given that suppressed IP3R3-mediated Ca$^{2+}$ flux and apoptosis has been linked to other major tumor suppressors PTEN, BAP1, and PML, in driving tumorigenesis[66–68], our above analyses suggest that this may also play a role in SMARCA4/2-deficient cancers. We examined this possibility in vivo using a xenograft model of H1703 cells with exogenous SMARCA4 expression, using a validated

doxycycline-controlled expression system[49]. Upon tumor establishment, we induced SMARCA4 expression with doxycycline treatment, which indeed resulted in suppression of tumor growth (Fig. 5i). Furthermore, IHC analysis of endpoint tumors showed that induced-SMARCA4 expression led to elevated expression of IP3R3 and cleaved caspase 3 (Fig. 5j, k). While this requires further studies, these data support that reduced IP3R3 expression in SMARCA4/2-deficient cancers may directly contribute to the tumorigenesis through suppression of apoptosis.

**Histone deacetylase inhibitor (HDACi) rescues IP3R3 expression and enhances cisplatin response in SMARCA4/2-deficient cancer cells.** Our data show that SMARCA4/2-deficient cancer cells are resistant to cisplatin in part through suppression of IP3R3 and that ectopic IP3R3 expression can sensitize these cancer cells to cisplatin-induced apoptosis (Fig. 4g–k, Supplementary Fig. 10). Although IP3R3 is not targetable, its expression is directly activated by SMARCA4/2 (Fig. 3). In contrast to deleterious mutations in *SMARCA4*, SMARCA2 loss is caused by epigenetic silencing in SCCOHT and NSCLC[17,71–73]. Furthermore, HDACi, a class of anti-cancer drugs that blocks the deacetylation of chromatin and other cellular substrates involved in cancer initiation and progression[74,75], has also been shown to reactivate SMARCA2 expression in SCCOHT and lung cancer cells[17,76,77]. Indeed, treatments with quisinostat[78], a second-generation HDACi, resulted in strong activation of SMARCA2 with concomitant elevation of IP3R3 at both mRNA and protein levels in SCCOHT (Supplementary Fig. 13) and SMARCA4/2-deficient NSCLC cancer cells (Fig. 6a, b). Consistent with this, quisinostat treatment strongly elevated cytosolic (Supplementary Fig. 14a) and mitochondrial (Supplementary Fig. 14b) Ca$^{2+}$ contents in response to histamine stimulation, similar to the levels induced by ectopic SMARCA4 expression in H1703 cells (Supplementary Fig. 14a–c). Notably, siRNA-mediated knockdown of IP3R3 in these quisinostat-treated cells prevented ER-Ca$^{2+}$ release, characterized by a significant decrease of cytosolic and mitochondrial Ca$^{2+}$ contents (Supplementary Fig. 14a–c). Together, these data indicate that quisinostat treatment can indirectly restore IP3R3 expression and rescue Ca$^{2+}$ flux in SMARCA4/2-deficient cancer cells.

Next, we explored the possibility of using HDACi to restore chemotherapy sensitivity in SMARCA4/2-deficient cancer cells. Indeed, the combination treatment of cisplatin and quisinostat in H1703 cells resulted in strong elevation of cleaved PARP and cleaved caspase 3 (Fig. 6b), the annexin V$^+$/PI$^-$ apoptotic cell population (Fig. 6c), and growth suppression (Fig. 6d). Given that HDACi is expected to activate expression of genes other than *SMARCA2*, it was important to verify the essential contribution of SMARCA2 reactivation to apoptosis induction by this drug

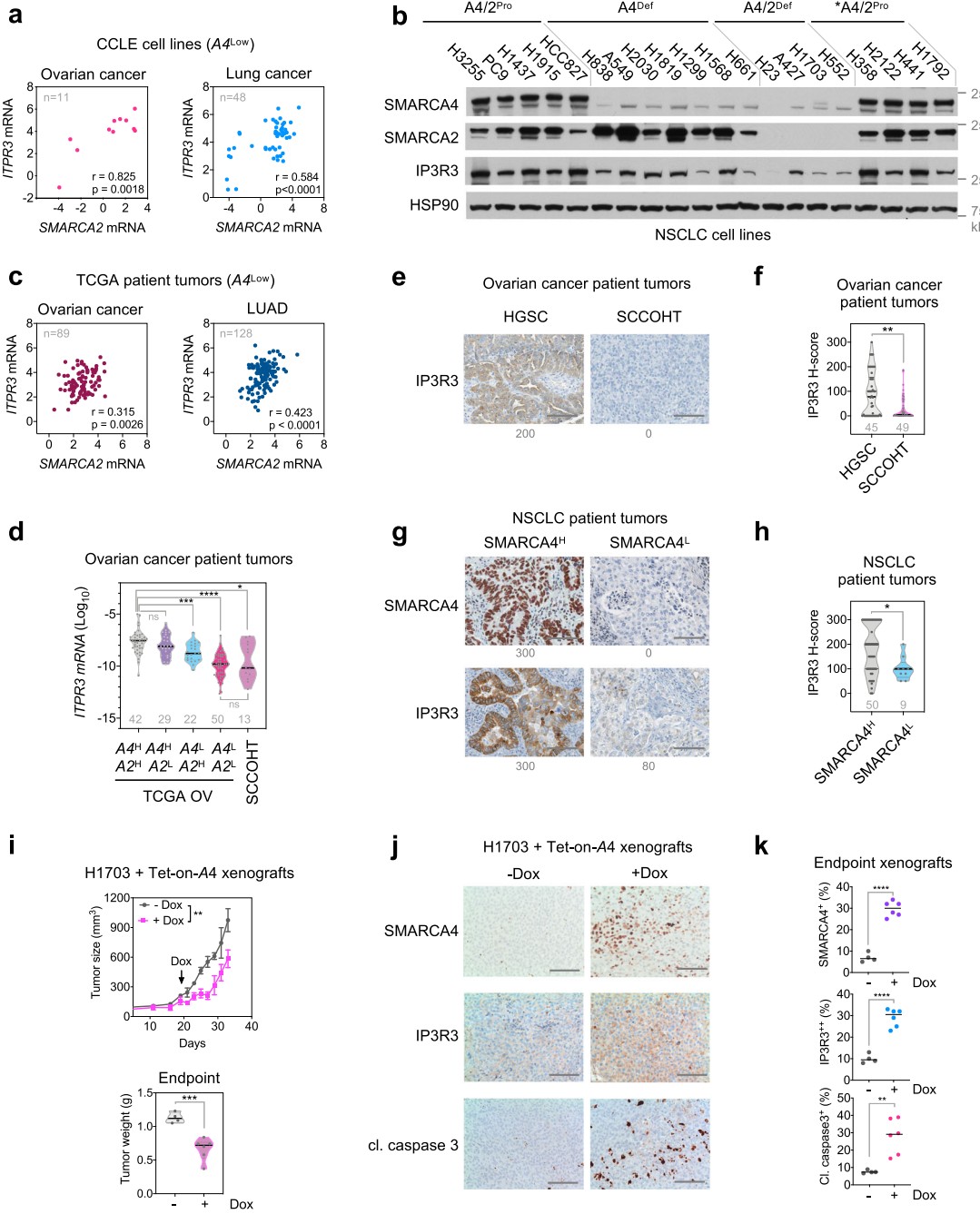

**Fig. 5 IP3R3 expression is reduced in SMARCA4/2-deficient cancers. a** Correlation of *ITPR3* and *SMARCA2* (*A2*) mRNA in ovarian (*n* = 11) and lung (*n* = 48) cancer cell lines with low expression of *SMARCA4* (*A4*). Expression data were obtained from Cancer Cell Line Encyclopedia (CCLE) in reads per kilobase million (RPKM)[42]. *A4*^Low, bottom quartile. *r*: Pearson correlation; *p*: *p* value (two-tailed). **b** Immunoblots of lung cancer cell lines with indicated SMARCA4/2 (A4/2) status. Pro proficient, def deficient; * *KRAS* mutant. **c** Correlation of *ITPR3* and *SMARCA2* mRNA in ovarian cancer (*n* = 89) and lung adenocarcinoma (LUAD, *n* = 128) patient tumors with low expression of *SMARCA4*. Gene expression data were obtained from UCSC Xena in fragments per kilobase million (FPKM). *A4*^Low, bottom quartile. *r*, Pearson correlation; *p*, *p* value (two-tailed). **d** *ITPR3* mRNA expression in SCCOHT and ovarian cancer patient tumors. TCGA ovarian cancers (OV) (*n* = 379) were stratified based on *SMARCA4/2* expression as indicated in Supplementary Fig. 2b. *ITPR3* expression in FPKM was normalized to *ACTB*. H: high; L: low. One-way ANOVA Brown–Forsythe and Welch tests followed by Dunnett's test for multiple comparisons to *A4*^H*A2*^H, *p* values (*p*): *A4*^H*A2*^L—0.3830, *A4*^L*A2*^H—0.0009, *A4*^L*A2*^L < 0.0001, SCCOHT—0.0108, or two-tailed *t*-test between *A4*^L*A2*^L group and SCCOHT, *p* = 0.4953. Representative images (**e, g**) and *H*-score (**f, h**) of immunohistochemistry analysis for IP3R3 and SMARCA4 expression in patient tumors. **e, f** HGSC (*n* = 49) and SCCOHT (*n* = 45). **g, h** NSCLC (*n* = 59). Scale bar, 100 μm. Mann–Whitney test (two-tailed), *p* values (*p*): **f** 0.0066, **h** 0.0270. **i** Tumor growth in H1703 xenograft models ± exogenous SMARCA4 expression. Doxycycline (Dox) was given daily starting on day 21 to induce SMARCA4. Upper, tumor size; lower, endpoint tumor weight. Mean ± SEM, − Dox (*n* = 4 animals), + Dox (*n* = 6 animals), two-way ANOVA (upper), two-tailed *t*-test (lower), *p* values (*p*): upper—0.0014, lower—0.0008. Representative images (**j**) and digital quantification (**k**) of immunohistochemistry analysis in endpoint tumors described in **i**. Scale bar, 100 μm. Mean ± SD, − Dox (*n* = 4), + Dox (*n* = 6), two-tailed *t*-test (lower), *p* values (*p*): upper, middle <0.0001; lower—0.0041. **p* < 0.05, ***p* < 0.01, ****p* < 0.001, *****p* < 0.0001; ns not significant.

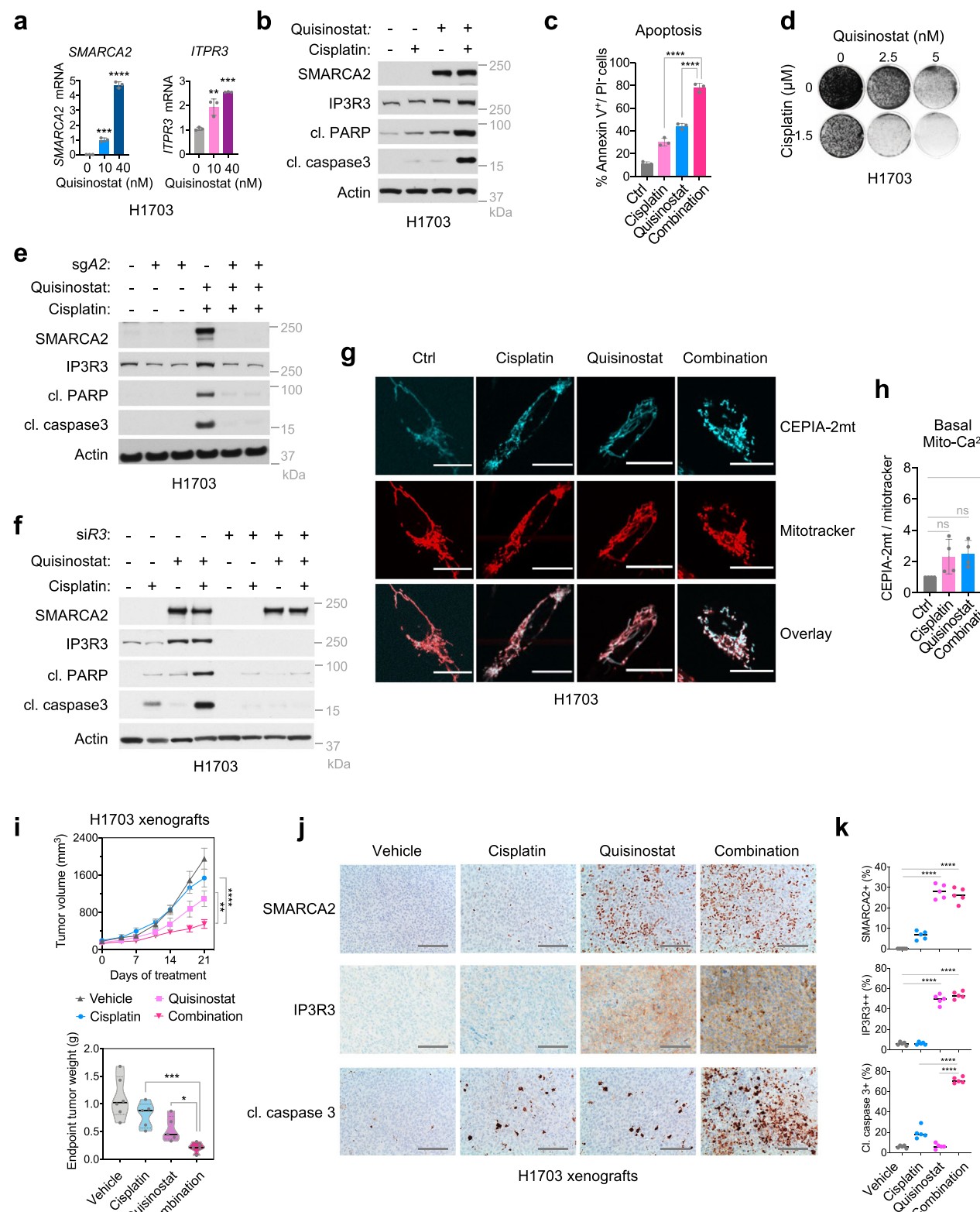

combination. Supporting this, CRISPR/Cas9-mediated *SMARCA2* knockout in H1703 cells blunted the elevation of IP3R3 and cell apoptosis markers induced by combination treatment of quisinostat and cisplatin (Fig. 6e). Furthermore, siRNA-mediated knockdown of IP3R3 also prevented the elevation of apoptosis markers in H1703 cells induced by this treatment combination (Fig. 6f). Finally, confocal live-cell imaging demonstrated that the combination of cisplatin and quisinostat strongly induced an increase of

basal mitochondrial $Ca^{2+}$ levels in these cells (Fig. 6g, h). These results demonstrate that quisinostat can stimulate SMARCA2-dependent IP3R3 expression to restore ER-$Ca^{2+}$ release-induced mitochondrial $Ca^{2+}$ flux and chemotherapy sensitivity in SMARCA4/2-deficient cancer cells.

Finally, we validated the antitumor effect of this cisplatin and quisinostat drug combination in vivo using a xenograft model of H1703 cells. After tumor establishment, animals were treated with

**Fig. 6 The histone deacetylase inhibitor quisinostat rescues IP3R3 expression and enhances cisplatin response in SMARCA4/2-deficient cancer cells.**
**a** *SMARCA2* and *ITPR3* mRNA expression in H1703 cells treated with quisinostat for 48 h. Mean ± SD, $n = 3$ independent experiments, one-way ANOVA followed by Dunnett's tests for multiple comparisons to untreated, $p$ values ($p$): left, 10 nM—0.0003, 40 nM < 0.0001; right, 10 nM—0.0027, 40 nM—0.0002. Immunoblots (**b**) and annexin V$^+$/PI$^-$ apoptotic population (**c**) in H1703 cells treated with cisplatin (3 µM) and quisinostat (10 nM) for 72 h. cl cleaved. Mean ± SD, $n = 3$ independent experiments, one-way ANOVA followed by Dunnett's test for multiple comparisons, $p$ values: all <0.0001.
**d** Colony formation assay for H1703 cells treated with cisplatin and quisinostat for 12 days. Immunoblots of H1703 cells ± SMARCA2 knockout (**e**) or ±IP3R3 knockdown (**f**) treated with cisplatin (3 µM) and quisinostat (10 nM) for 72 h. **g** Representative images from confocal live-cell imaging of H1703 cells overexpressing the mitochondrial Ca$^{2+}$ probe CEPIA-2mt treated with quisinostat or/and cisplatin and stained with the mitochondrial marker Mitotracker deep red. Cisplatin: 2 µM, 24 h; quisinostat: 40 nM, 72 h. Scale bar, 25 µm. **h** Quantification of basal mitochondrial Ca$^{2+}$ levels from **g**, showing the ratio of CEPIA-2mt/Mitotracker fluorescence intensities compared to control. 43 control (ctrl), 42 cisplatin, 38 quisinostat, and 46 quisinostat/cisplatin cells from $n = 4$ independent experiments were analyzed. Mean ± SD, one-way ANOVA followed by Dunnett's test for multiple comparisons to ctrl, $p$ values: cisplatin—0.2105, quisinostat—0.2985, combination—0.0032. **i** Tumor growth in H1703 xenograft models treated with cisplatin (4 mg kg$^{-1}$) and quisinostat (10 mg kg$^{-1}$). Vehicle ($n = 6$ animals), all other groups ($n = 5$ animals); upper, tumor volume, mean ± SEM, two-way ANOVA; lower, endpoint tumor weight, one-way ANOVA followed by Dunnett's tests for multiple comparisons to the combination, $p$ values: upper, cisplatin < 0.0001, quisinostat—0.0054; lower, cisplatin—0.0008, quisinostat—0.0330. Representative images (**j**) and digital quantification (**k**) of immunohistochemistry analysis in endpoint tumors described in **i**. Scale bar, 100 µm. Mean ± SD, all groups ($n = 5$), one-way ANOVA followed by Dunnett's test for multiple comparisons, $p$ values: all <0.0001. *$p < 0.05$, **$p < 0.01$, ***$p < 0.001$, ****$p < 0.0001$; ns not significant.

cisplatin (4 mg kg$^{-1}$), quisinostat (10 mg kg$^{-1}$), or their combination. Consistent with our in vitro results, the combination more effectively suppressed tumor growth than each single drug alone, as indicated by a significant reduction of both tumor volume and weight (Fig. 6i). We noted that some animals treated with cisplatin or the combination, but not quisinostat alone, showed body weight loss (Supplementary Fig. 15a), likely associated with chemotherapy-induced side effects. Nevertheless, when normalized to the animal body weight, the drug combination still showed significant reduction of both tumor volume and weight compared to single treatments (Supplementary Fig. 15b, c). Furthermore, IHC analysis of endpoint tumors revealed that quisinostat treatment was able to induce protein expression of SMARCA2 and IP3R3 (Fig. 6j, k) and, when combined with cisplatin, synergistically elicited a strong apoptotic response as indicated by a marked increase of cleaved caspase 3 levels (Fig. 6j, k). Taken together, our data provide a proof-of-concept treatment strategy for enhancing chemotherapy response in patients affected by SMARCA4/2-deficient cancers.

## Discussion

We show that SMARCA4/2 deficiency impairs chemotherapy-induced apoptotic responses in ovarian and lung cancers at least in part by altering ER to mitochondria Ca$^{2+}$ flux. By directly restricting *ITPR3* expression, SMARCA4/2 loss inhibits Ca$^{2+}$ transfer from the ER to mitochondria required for apoptosis induction. Consequently, stimulation of *ITPR3* expression through SMARCA2 reactivation by HDACi enhanced chemotherapy response in SMARCA4/2-deficient cancer cells.

SWI/SNF subunits are frequently mutated in human cancers[4], which has been connected to hallmarks of cancers including aberrant cell proliferation, lineage differentiation, and altered metabolism[1,28]. Our findings establish a functional link between SMARCA4/2 loss and dampened IP3R3-mediated Ca$^{2+}$ flux in resisting programmed cell death. While our current study mostly focuses on chemoresistance, we also found that SMARCA4 restoration alone suppressed tumor growth of H1703 xenografts associated with increased expression of IP3R3 and cleaved caspase 3. This suggests that altered Ca$^{2+}$ homeostasis may also directly contribute to the tumorigenesis of SMARCA4/2 loss through suppression of apoptosis, as previously shown for other major tumor suppressors PTEN, BAP1, and PML[66–68]. Additional investigations are warranted to further confirm these results. Given the cooperative roles of SMARCA4/2 in regulating ER to mitochondria Ca$^{2+}$ flux and apoptosis, it is likely that they exert these functions in a SWI/SNF-dependent manner. Therefore, exploring the potential role of other SWI/SNF subunits

frequently altered in cancers, such as ARID1A[4], in Ca$^{2+}$ homeostasis and apoptosis may help understand the oncogenic mechanisms underlying other SWI/SNF-deficient cancers.

Our study examined the roles of SMARCA4/2 in regulating chemotherapy response and apoptosis induction using cancer cell lines that naturally harbor SMARAC4/2 alterations. This is different from previous studies employing RNAi-mediated *SMARCA4* knockdown in SMARCA4-proficient cancer cells which led to enhanced response to DNA damaging agents[31–33]. We found that naturally occurred SMARCA4/2-deficient cancer cells are more resistant to chemotherapy, which is in line with previous reports showing that SCCOHT is typically more resistant to conventional chemotherapy in both cell models and patients[15,34]. Similarly, experimental suppression of SMARCA2 has been shown to be selective lethal to SMARCA4-deficient/SMARCA2 proficient cancer cells[18–20]. However, concomitant loss of SMARCA4/2 occurs in almost all SCCOHTs and a subset of NSCLCs associated with poorer prognosis in patients[6,10,16,17]. Therefore, naturally occurred SMARC4A/2-deficient cancers may represent a unique group with distinct properties such as altered Ca$^{2+}$ homeostasis leading to chemotherapy resistance.

Similar to SCCOHT, our analysis in multiple NSCLC datasets of diverse tumor staging including the most comprehensive Director's Challenge data set suggests that reduced *SMARCA4* expression is associated with chemoresistance in NSCLC. A previous report[79] analyzing the JBR.10 data set of NSCLCs from early stages[80] showed that patients whose tumors expressed low *SMARCA4*, but not high *SMARCA4*, benefited from the adjuvant therapy of cisplatin and vinorelbine (a microtubule inhibitor). This discrepancy is likely due to differences in *SMARCA4* microarray probe sets chosen, patient cohort compositions, and data analysis methods. While we used the optimal "Jetset probe" unbiasedly identified by the KM plotter without preassociation with patient outcome, microarray technology has limited sensitivity and specificity in quantifying gene expression. Thus, these results require further confirmation using better tools such RNA-seq. In addition, we recognize that patient outcome is often influenced by multiple factors such as treatment history, which was not uniform among all patients analyzed. Therefore, additional clinical studies are needed to better control these variants and evaluate roles of SMARCA4/2 expression in predicting chemotherapy responses in NSCLC patients.

HDACi have been clinically approved for the treatment of several hematological malignancies but their activity in solid tumors has been limited as single agents[74,75]. Thus, identifying genetic vulnerability of HDACi and effective drug combinations

may enhance their clinical utility. SCCOHT cells have been shown to be more sensitive to HDACi than SMARCA4/2-deficient NSCLC cells[76]. This may be because NSCLC have a more complex genetic make-up than SCCOHT[15,62]. Our study provided proof-of-principle data supporting that HDACi may be a potential therapeutic strategy to stimulate *ITPR3* transcription through SMARCA2 reactivation and sensitize SMARCA4/2-deficient cancers to chemotherapy. Other strategies may also be explored. For example, GGTi-2418, a geranylgeranyl transferase inhibitor, sensitizes A549 cells to apoptosis induction by photo-dynamic therapy both in vitro and in xenograft models via stabilizing the IP3R3 protein[66]. Of note, A549 is also a SMARCA4-deficient NSCLC cell line and this independent study does further support the notion of elevating IP3R3 expression to enhance chemotherapy response in SMARCA4/2-deficient cancers. However, both HDACi and GGTi-2418 intervene IP3R3 expression indirectly and may cause unexpected toxicity. Therefore, other agents that directly facilitate $Ca^{2+}$ flux from the ER to mitochondria need to be investigated in the future. In addition to IP3R3, other common targets of SMARCA4/2 may also play a role in altered $Ca^{2+}$ homeostasis impacting apoptosis, which could serve as potential drug targets in SMARCA4/2-deficient cancers and will require further studies.

In summary, we have uncovered that SMARCA4/2 loss restricts IP3R3-mediated $Ca^{2+}$ flux from the ER to mitochondria, leading to resistance to chemotherapy-induced apoptosis in ovarian and lung cancers. Our study provides insights into the molecular mechanisms of SWI/SNF loss in promoting drug resistance and suggests a potential therapeutic strategy to enhance chemotherapy response in patients affected by SMARCA4/2-deficient cancers.

## Methods

**Cell culture.** All cell lines were cultured in Roswell Park Memorial Institute 1640 Medium (Thermo Fisher Scientific, Cat# 11875-093) with 7% fetal bovine serum (Sigma, Cat# F1051), 1% penicillin/streptomycin (Thermo Fisher Scientific, Cat# 15140-122), and 2 mM L-glutamine (Thermo Fisher Scientific, Cat# 25030-081), except for 293T with Dulbecco's modified Eagle medium (Thermo Fisher Scientific, Cat# 11995-065) and HEC116 with Eagle's minimum essential medium (Wisent, Cat# 320-005-CL). Cells were maintained at 37 °C and 5% $CO_2$ and regularly tested for Mycoplasma using Mycoalert Detection Kit (Lonza, Cat # LT07-318). All cell line origins are listed in Reporting Summary and have been validated by short tandem repeat analysis.

**Lentivirus production and infection.** All experiments with ectopic expression, shRNA knockdown, and CRISPR single guide RNA (sgRNA) knockout were performed using lentiviral constructs. For lentivirus production, $2.5 \times 10^6$ 293T cells were plated in 2 mL of DMEM medium per well in a six-well plate and transfected after ~8 h with lentiviral constructs, the packaging (psPAX2), and envelope plasmid (pMD2.G) by $CaCl_2$. Virus containing medium were harvested at 24 and 36 h after transfection before use or stored at −80 °C. For infection, ~$5 \times 10^5$ target cells were plated the day before and infected with virus for ~8 h. After ~20 h recovery, cells were selected in medium containing 2 µg/mL puromycin or 5 µg/mL blasticidin for 2–3 days and harvested for the experiments.

**Compounds and antibodies.** Cisplatin (S1166), quisinostat (S1096), paclitaxel (S1150), and topotecan (S9321) were purchased from Selleck Chemicals. Cyclophosphamide (CA80500-080), histamine (H7125), and thapsigargin (T9033) were from Sigma-Aldrich. Mitotracker deep red FM was from Thermo Fisher Scientific (M22426). Antibodies against calregulin (Cat# sc-166837), HSP90 (Cat# sc-13119), and β-Actin (Cat# sc-47778) were from Santa Cruz Biotechnology; antibodies against SMARCA2 (Cat# 11996), cleaved PARP (Cat# 5625), and cleaved caspase 3 (Cat# 9664) were from Cell Signaling; antibodies against MICU2 (Cat# ab-101465), VDAC1 (Cat# ab-14734), and GRP75 (Cat# ab-2799) were from Abcam; antibody against SMARCA4 (Cat# A300-813A) was from Bethyl Laboratories (A300-813A); antibody against IP3R3 (Cat# 610312) was from BD Pharmingen; antibody against vinculin (Cat# V4505) was from Sigma-Aldrich; antibody against MCU (Cat# HPA0168480) was from Atlas; and antibody against MICU1 (Cat# orb-323178) was from Biorbyt. Antibody against SMARCA4 was used with 1:5000 dilution and all others with 1:1000 dilution. Antibodies for IHC are described in the Immunohistochemistry method section below. Secondary antibodies goat anti-mouse IgG (Cat#1706516) and goat anti-rabbit IgG (Cat#1706515) were purchased from Bio-Rad.

**Plasmids.** Individual shRNA vectors used were from the Mission TRC library (Sigma) provided by McGill Platform for Cellular Perturbation (MPCP) of Rosalind and Morris Goodman Cancer Research Centre and Biochemistry at McGill University: sh*SMARCA2*#1 (TRCN0000358828); sh*SMARCA2*#2 (TRCN0000020333); sh*ITPR3*#1 (TRCN0000061324); and sh*ITPR3*#2 (TRCN0000061326). For shRNA experiments, pLKO vector control was used. Scramble control sgRNA (SCR_6) and dual-sgRNAs targeting *SMARCA4* (TEDH-1074701) or *SMARCA2* (TEDH-1074696) were from the transEDIT-dual CRISPR Library (Transomic) provided by MPCP. Additional sgRNA (GCTGGCCGAG-GAGTTCCGCCC) targeting *SMARCA4* was cloned into pLentiCRISPRv2. pReceiver control vector, pReceiver-*SMARCA4*, and pReceiver-*SMARCA2* were purchased from GeneCopoeia. pLX304-*ITPR3* was generated by gateway cloning with pENTR223.1-*ITPR3* (BC172406). pENTR223.1-*ITPR3* (BC172406) and pLX304-*GFP* control (ccsbBroad304_07515) were from Transomic provided by MPCP. transEDIT™ pCLIP-All-EFS-Puro Epigenetics CRISPR Screening library was from Transomic (Cat# CAHD9001). pLentiCas9-Blast (Cat# 52962), pLenti-CRISPRv2 (Cat# 52961), pCMV-R-GECO1 (Cat# 32444), and pCMV-CEPIA2mt (Cat# 58218) were from Addgene. pIN20 and pIN20-SMARCA4[82] were provided by Dr. Jannik N. Andersen (The University of Texas, MD Anderson Cancer Center).

**CRISPR/Cas9 editing.** Plasmid-based CRISPR/Cas9 editing was used to generate SMARCA4 knockout in OVCAR4 and H1437 cells using standard lentiviral delivery followed by single-cell cloning. For HEC116 cells, ribonucleoprotein (RNP) delivery was used. cRNA targeting SMARCA4 (sequence = GCGGTGGCATCACGGGCG) and tracrRNA duplexes (1 µM) were formed by heating at 95 °C and gradual cool down to room temperature (RT). RNP complexes were formed by combining the 1 µM guide RNA oligos with 1 µM Alt-R *S. pyogenes* Cas9 endonucleases (IDT) in Gibco Opti-MEM media (Thermo Fisher Scientific) for 5 min at RT. Transfection complexes containing the RNP complex and Lipofectamine RNAiMAX transfection reagent (Thermo Fisher Scientific) were diluted in Opti-MEM media and incubated at RT for 20 min. HEC116 endometrial cancer cells were added to transfection complexes in the wells of a 24-well tissue culture plate to achieve a final concentration of 40,000 cells/well and final concentration of RNP of 10 nM. Flow cytometry (University of Alberta, Faculty of Medicine and Dentistry, Flow Cytometry Facility) was utilized to enrich for CRISPR transfected cells positive for tracrRNA-ATTO™ 550 fluorescence. Single clones were either generated by flow cytometry plating a single cell per well into a 96-well plate or manually plating of 0.5 cells/well into a 96-well plate upon filtration through a cell strainer. Single-cell-derived clones were validated by Sanger sequencing over the guided nuclease target region.

**CRISPR sgRNA screen.** The transEDIT™ pCLIP-All-EFS-Puro Epigenetics CRISPR Screening library (Transomic) containing 5080 sgRNAs targeting 496 epigenetic genes was used in this study. Library virus was generated in 293T cells as described above. OVCAR4 cells were infected with library virus at low multiplicity of infection achieving single sgRNA integration. After selection, ~$5 \times 10^6$ cells (1000-time coverage) were plated in 15 cm dishes and treated with ±100 nM cisplatin for 14 days before harvesting. Genomic DNA was isolated with High Pure PCR Template Preparation Kit (Roche) by following the manufacturer's instruction. Library preparation for next-generation sequencing was done as described before[83]. Briefly, the gRNA sequences were amplified from 20 µg genomic DNA by PCR using Phusion HF DNA polymerase (Thermo Fisher Scientific) using a two-step amplification adding a unique 6-bp index per sample and sequencing adapter sequences. PCR products were purified using the High Pure PCR Product Purification Kit (Roche) and quantified using the Quant-iT™ PicoGreen™ dsDNA Assay Kit (Thermo Fisher Scientific) before sequencing on a HiSeq2500 System (Illumina). Sequencing reads were mapped to the library using xcalibr and counts were then analyzed with MAGeCK (version 0.5.8) using the robust rank aggregation algorithm. Primers used are as follows: PCR1: PTRC_index (forward), IllSeqR_CR_r (reverse); PCR2: P5_Illuseq (forward), P7_index_IR_r (reverse). Please see Supplementary Table 3 for primer sequences. Index sequences: Control: ACATCG, cisplatin: GCCTAA. Please see Supplementary Table 3 for sequence details.

**Colony formation assays.** Since different cell lines have variable proliferation rates and sizes, plating densities for each line were first optimized to allow about two weeks of drug treatment, before cells reach 90% confluency in six-well plates. Single-cell suspensions of all cell lines were then counted and seeded into six-well plates with the densities predetermined ($2–8 \times 10^4$ cells/well). Cells were treated with vehicle control or drugs on the next day and culture medium was refreshed every 3 days for 10–14 days in total. At the endpoints of colony formation assays, cells were fixed with 3.75% formaldehyde, stained with crystal violet (0.1%w/v), and photographed. All relevant assays were performed independently at least three times.

**Cell viability assays.** Cultured cells were seeded into 96-well plates (1,000–6,000 cells per well). Twenty-four hours after seeding, different dilutions of compounds were added to cells. Cells were then incubated for 4 days and cell viability was

measured using the CellTiter-Blue Viability Assay (Promega) by measuring the fluorescence (560/590 nm) in a microplate reader. Relative survival in the presence of drugs was normalized to the untreated controls after background subtraction.

**Protein lysate preparation and immunoblots**. Cells were first seeded in normal medium without inhibitors. After 24 h, the medium was replaced with fresh medium containing the inhibitors as indicated in the text. After the drug stimulation, cells were washed with cold PBS, lysed with protein sample buffer, and processed with Novex® NuPAGE® Gel Electrophoresis Systems (Thermo Fisher Scientific). HSP90, actin, vinculin, or calreticulin were used as loading controls. The quantification of immunoblots was performed on two independent experiments using Image J and normalized to loading controls are displayed in Supplementary Fig. 16. Uncropped immunoblots presented in main figures are displayed in Supplementary Fig. 17. Uncropped immunoblots presented in Supplementary Figures are included in Source Data.

**Annexin V staining and IncuCyte imaging**. Cells in 96-well plates were treated with cisplatin at different concentrations in medium containing IncuCyte® annexin V for apoptosis (Essen Bioscience, Cat# 4641). IncuCyte® live-cell analysis imaging system was used to record four images every 2–4 h. Images were analyzed by IncuCyte® Zoom (2016B) software and annexin V positive cells were normalized to phase-contrast confluency for each well.

**Annexin V and PI flow cytometry**. For apoptosis assays, negative controls were prepared by incubating cells in the absence of inducing agent and positive controls for apoptosis were prepared using 10 μM $H_2O_2$. Cells were harvested after treatment and washed in cold phosphate buffered saline and resuspended in 1X annexin binding buffer (BMS500BB) to $10^6$ cells/mL. One hundred microliters of cell suspension was used per assay and 5 μL of FITC annexin V (A13199) and 1 μL of PI (P1304MP) diluted to 100 μg/mL in annexin V binding buffer was added to each suspension. Cells were incubated following addition of fluorescent reagents for 15 min at RT. Four hundred microliters of 1X annexin V binding buffer were added to each suspension following incubation and the samples were mixed gently and kept in the dark and on ice prior to analysis.

Flow cytometry was performed using Guava easyCyte HT (Sigma) with the guavaSoft 2.5 software (Sigma) based on the manufacturer's instructions. Fluorescence emission was measured at 530 and >575 nm to separate between the annexin $V^+$ and $PI^+$ populations in green and red. Technical controls for gating were prepared with uninduced cells with both PI and annexin V stains, with either PI or annexin V only, or in the absence of both. Apoptotic cell population (annexin $V^+/PI^-$) showed green fluorescence only. Gating strategy is exemplified in Supplementary Fig. 18.

**RNA isolation and real-time quantitative reverse transcription PCR (qRT-PCR)**. After indicated drug treatment or genetic modifications, cells were harvested for RNA isolation using Trizol (Thermo Fisher Scientific, Cat # 15596018) the next day. Synthesis of complementary DNAs (cDNAs) using Maxima First Strand cDNA Synthesis Kit (Thermo Fisher Scientific, Cat# K1642) and qRT-PCR assays using PowerUp™ SYBR™ Green Master Mix (Thermo Fisher Scientific, Cat# A25742) were carried out according to the manufacturer's protocols. Relative mRNA levels of each gene shown were normalized to the expression of the housekeeping gene *GAPDH*. The sequences of the primers are listed in Supplementary Table 3.

**Survival analysis**. Survival analyses were performed on LUAD patients from the following datasets with available information on adjuvant chemotherapy status: Director's Challenge Consortium for the Molecular Classification of Lung Adenocarcinoma[36], KM plotter[37], and The UT Lung SPORE (GSE42127)[39]. For all Affymetrix microarray datasets, 213720_s_at was the probe used to assess *SMARCA4* expression. Li et al. defined this probe as the "JetSet" probe—the most suitable gene probe based on its specificity, coverage, and degradation resistance characteristics[38]. Director's Challenge and The UT Lung SPORE (GSE42127) datasets were analyzed by stratifying patients into *SMARCA4* high and low groups, separated by median *SMARCA4* level. The survival data were analyzed by one-tailed Mantel–Cox analysis in GraphPad Prism. Parameters for kmplot.com query were: gene symbol—*SMARCA4*; probe set options—use JetSet best probe set; split patients by—auto select best cutoff; survival—censor at threshold; histology—adenocarcinoma; and all other default settings. Patients with and without adjuvant chemotherapy were analyzed separately in all datasets.

**Transcriptome analysis**

*Cell lines*. There were three sets of transcriptome data used in this study, namely SMARCA4 restoration in BIN-67 and SCCOHT-1 cells (GSE120297, RNA-seq), SMARCA4/2 restoration in BIN-67 cells (GSE117735, RNA-seq), and SMARCA4 restoration in H1299 cells (GSE109010, microarray). Processed gene expression data were retrieved from original study for GSE120297 and by GEOquery (2.56.0)[84] for GSE109010. For GSE117735, sequencing files were downloaded from sequence read archive and mapped to reference human genome sequence (hg19)

with STAR (2.6.1c)[85]. Gene expression counts were calculated by featureCounts (v1.6.4)[86] with UCSC hg19 gene annotation GTF file. Heatmaps for gene expression were generated with pheatmap (1.0.12) after normalization. Differential expression genes were identified with DESeq2 (version 1.19.38) for GSE120297, with GEO2R analysis for GSE109010 and from original study[60] for GSE117735.

*Patient tumors*. Total RNA from three SCCOHT patient tumors was subjected for RNA-seq at Genome Quebec. Briefly, quantification was performed using a NanoDrop Spectrophotometer ND-1000 (NanoDrop Technologies, Inc.) and its integrity was assessed using a 2100 Bioanalyzer at Genome Quebec. Libraries were generated from 250 ng of total RNA using the TruSeq stranded mRNA Sample Preparation Kit (Illumina, Cat# RS-122-2101), as per the manufacturer's recommendations. Libraries were quantified using the Quant-iT™ PicoGreen® dsDNA Assay Kit (Thermo Fisher Scientific, Cat# P7589) and the Kapa Illumina GA with Revised Primers-SYBR Fast Universal kit (Kapa Biosystems). Average size fragment was determined using a LabChip GX (PerkinElmer) instrument. RNA-seq data of another ten SCCOHT patient tumors were obtained from a previous study[87]. Sequencing results were processed by following mRNA quantification analysis pipeline of Genomic Data Commons (https://docs.gdc.cancer.gov/Data/ Bioinformatics_Pipelines/Expression_mRNA_Pipeline/): first aligning reads to the GRCh38 reference genome with STAR-2.6.0c and then by quantifying the mapped reads with HTSeq-0.6.1[88]. RNA-seq read counts of 379 ovarian cancer tumors were obtained from UCSC Xena (http://xena.ucsc.edu/) which followed the exact same pipeline. The fragments per kilobase of transcript per million mapped reads (FPKM) for each gene was calculated as

$$FPKM = (RC_g \times 10^9)/(RC_{pc} \times L)$$

where $RC_g$ is the number of reads mapped to the gene; $RC_{pc}$ is the number of reads mapped to all protein-coding genes; and $L$ is mean of lengths of isoforms of a gene.

**Gene set enrichment analysis**. Preranked gene listed were generated on the $\log_2$ transformed fold change for significantly changed genes (adjusted $p$ value smaller than 0.05). The R package clusterProfiler (v3.12.0)[89] was used to perform GSEA with the following parameters: ont = "MF", nPerm = 10,000, minGSSize = 3, maxGSSize = 800, and pvalueCutoff = 0.05.

**siRNA and plasmids transfection**. For siRNA experiments, cells were transfected using Lipofectamine RNAimax (Thermo Fisher Scientific, Cat# 13778150) with 20 nM SMARTPool ON-TARGETplus HUMAN *ITPR3* siRNA (Horizon Discovery, cat# L-006209-00-0005) for 3 days according to the manufacturer's recommendations. Plasmids were transfected for 24 h using FuGENE HD (Promega, Cat# E2311) following manufacturer's recommendations.

**Intracellular $Ca^{2+}$ measurements**. To measure cytosolic or mitochondrial $Ca^{2+}$, OVCAR4, H1703, SCCOHT-1, and H1437 cells were cultured on Nunc Lab-Tek chambered eight-well cover glass (Thermo Fisher Scientific, cat# 154534) and transiently transfected with the cytosolic R-GECO1 (Addgene, cat# 32444)[56] or mitochondrial CEPIA-2mt (Addgene, cat# 58218)[57] $Ca^{2+}$ reporter probes. Cells were washed three times in a balanced salt solution buffer + $Ca^{2+}$ (BSS) (120 mM NaCl, 5.4 mM KCl, 0,8 mM $MgCl_2$, 5.6 mM $NaHCO_3$, 5.6 mM D-glucose, 2 mM $CaCl_2$, and 25 mM HEPES [pH 7.3]). Fluorescence values were then collected every 2 s for 3 min. ER-$Ca^{2+}$ release was stimulated by injection of 100 μM histamine final in BSS + $Ca^{2+}$ at 10 s. Images were acquired using a 40× objective of the Nikon Eclipse Ti-E microscope, coupled to an Andor Dragonfly spinning disk confocal system equipped with an Andor Ixon camera, exciting with 488 nm or 561 nm laser for CEPIA-2mt or R-GECO1, respectively.

To measure total ER-$Ca^{2+}$ content, OVCAR4, H1703, SCCOHT-1, and H1437 cells were cultured on Nunc Lab-Tek chambered eight-well cover glass and transiently transfected with the cytosolic R-GECO $Ca^{2+}$ reporter probe. Cells were washed three times in a BSS-$Ca^{2+}$ (120 mM NaCl, 5.4 mM KCl, 0.8 mM $MgCl_2$, 6 mM $NaHCO_3$, 5.6 mM D-glucose and 25 mM HEPES [pH 7.3]). Fluorescence values were then collected every 2 s for 5 min. ER-$Ca^{2+}$ release was stimulated by injection of 10 μM thapsigargin final in BSS-$Ca^{2+}$ at 10 s. Images were acquired using microscope and laser described above.

To measure basal mitochondrial $Ca^{2+}$ pools, H1703 cells were cultured on Nunc Lab-Tek chambered eight-well cover glass, treated with appropriate drugs, and transiently transfected with the mitochondrial CEPIA-2mt $Ca^{2+}$ reporter probe. Cells were stained with 100 nM Mitotracker deep red (Thermo Fisher Scientific, cat# M22426) for 20 min followed by three washes in complete culture media prior to imaging. Fluorescence values were then collected every 2 s for 30 s. Images were acquired using a 40× objective of the Nikon Eclipse Ti-E microscope, coupled to an Andor Dragonfly spinning disk confocal system equipped with an Andor Ixon camera, exciting with 488 and 647 nm lasers for CEPIA-2mt and Mitotracker deep red, respectively.

**Immunohistochemistry**. Tissue microarrays (TMAs) of tumor samples of HGSC and NSCLC patients used in this study were previously described[48,49]. A TMA of 52 SCCOHT patient tumors was constructed for this study. Studies on SCCOHT patient tumors were approved by the Institutional Review Board (IRB) at McGill

University, McGill IRB # A08-M61-09B. Studies on 59 pathologist-confirmed ovarian HGSC samples were approved by the ethics boards at the University Hospitals Network and the Jewish General Hospital respectively. Informed consent was obtained from all participants in accordance with the relevant IRB approvals. Hematoxylin and eosin-stained sections of the 50 SCCOHTs (confirmed by DNA mutation analysis or/and SMARCA4 IHC) and 52 HGSC cases were reviewed and areas containing tumor only were demarcated and cored to construct TMAs using duplicate 0.6-mm cores from the demarcated areas. A panel of 100 resected LUAD patient tumors were analyzed. This study was approved by the ethics boards at the McGill University Health Centre (F11HRR, 17212). The NSCLC TMA was comprised of 4 mm cores from the selected paraffin-embedded tissue blocks. For all IHC analysis, cores with low tumor cellularity and artifacts were not included in the analysis.

The 4 µm thick sections from these TMAs were cut, deparaffinized, and stained using the BenchMark Ultra system (Ventana Medical Systems Inc). Heat-induced epitope retrieval (HIER) was performed with Ultra Cell Conditioning Solution (CC1) for 64 min at 95 °C, followed by 16 min of incubation at 36 °C with the rabbit monoclonal antibody against SMARCA4 (clone EPNCIR111A; dilution, 1:100; Abcam). For IP3R3, HIER was performed in CC1 for 48 min at 100 °C followed by incubation for 48 min at 36 °C with the mouse anti-IP3R3 (BD Transduction Laboratories). After primary antibody incubation, detection was performed using the default OptiView DAB protocol as per the manufacturer's directions (Ventana). The slides were digitalized using an Aperio scanner and the Lumenera INFINITY X CMOS camera.

For patient tumors, assessment of SMARCA4, unequivocally absent staining in the nuclei of viable tumor cells as opposed to strong staining in background stromal cells was considered IHC negative. Expression in the tumor cells that is equivalent to the staining of nonneoplastic cells in the background was considered IHC positive. Positive cells were analyzed according to the staining intensity on a scale of 0–3 (0 = negative, 1 = weak, 2 = moderate, 3 = strong). $H$-scores were calculated as the sum of the percent of cells at each intensity ($Pi$) multiplied by the intensity score ($i$). $H$-score = $\Sigma (Pi(i)) \times 100$. Score values range between 0 and 300. Cores with low tumor cellularity and artifacts were not included in the analysis.

For xenograft tumor sections, quantification of percentage positive staining for SMARCA4, SMARCA2, and cleaved caspase 3 was performed unbiasedly using the Aperio nuclear algorithm on Aperio ImageScope. Quantification of percentage positive staining for IP3R3 was performed unbiasedly using the Aperio cytoplasm algorithm on Aperio ImageScope. Weak IP3R3 (+) staining resulting from the background was not considered in the analysis.

**Mouse xenografts and in vivo drug studies.** Animal experiments were performed according to standards outlined in the Canadian Council on Animal Care Standards (CCAC) and the Animals for Research Act, R.S.O. 1990, Chapter c. A.22, and by following internationally recognized guidelines on animal welfare. All animal procedures (Animal Use Protocol) were approved by the Institutional Animal Care Committee according to guidelines of the CCAC. All animal experiments were carried out at the Goodman Cancer Research Center of McGill University, using 8–12-week-old in house bred male NOD.Cg-Prkdc$^{scid}$ Il2rg$^{tm1Wjl}$/SzJ (NSG) mice.

For in vivo drug studies, Quisinostat (SelleckChem) was resuspended in 20% hydroxypropyl-β-cyclodextrin (Sigma-Aldrich) buffer (pH = 8.70) at a concentration of 50 mg mL$^{-1}$ (administrated intraperitoneally at 10 mg kg$^{-1}$ dose for a 25–28 g mouse in a volume of 100 µL). Cisplatin (SelleckChem) was resuspended in 0.9% sodium chloride solution (administrated intraperitoneally at 4 mg kg$^{-1}$ dose for a 25–28 g mouse in a volume of 200 µL). These two reagents were stored at −80 °C. Tubes were thawed overnight at 4 °C.

For the tumor model, single-cell suspension was created by dissociating a sufficient number of subconfluent flasks of cells to produce $4 \times 10^6$ cells (H1703 or H1703 expressing pIN20-SMARCA4) in 200 µL of Matrigel HC in a 50:50 ratio (Corning Matrigel HC, VWR). The tumor cell suspension was subcutaneously injected into the left flank of each NSG mouse. For the doxycycline inducible model using H1703 cells expressing pIN20-SMARCA4, as tumor volumes ($V = (H \times W^2)/2$) reached ~150 mm$^3$ (21 days post inoculation), experimental mice ($n = 6$) were injected with 2.5 mg/mL doxycycline (Millipore Sigma) intraperitoneally followed by 2 mg/mL in sucralose (MediDrop, ClearH20) solution ad libitum. Experimental mice were again injected intraperitoneally with doxycycline at day 32 to ensure they were acquiring adequate drug. Control mice ($n = 4$) received intraperitoneal injections of saline (diluent) and received sucralose ad libitum. All mice were placed on sucralose a week prior to the experiment to acclimatize mice to the taste. For the chemo drug treatment experiment, when tumor volumes reached ~150 mm$^3$ (20 days post inoculation), which was assigned as day 0, the mice were entered into the treatment regimen (21 days). Mice were randomly allocated to control (vehicle, $n = 6$), quisinostat (10 mg kg$^{-1}$ quisinostat[76], three times per week, $n = 5$), cisplatin (4 mg kg$^{-1}$ cisplatin[90], once per week, $n = 5$), or combination (10 mg kg$^{-1}$ quisinostat and 4 mg kg$^{-1}$ cisplatin, $n = 5$) group. Mice were housed in groups of 4–5, with each group consisting of both vehicle control and treatment animals matched for tumor size on day 0 of treatment. Tumor progression was monitored and measurements using digital calipers (VWR) were recorded twice weekly. The persons who performed all the tumor measurements and the IHC analysis for the endpoint tumor samples were blinded to the treatment information.

**Statistics and reproducibility.** GraphPad Prism 8 software was used to generate graphs and statistical analyses. Methods for statistical tests, exact value of $n$, and definition of error bars are indicated in figure legends, *$p < 0.05$, **$p < 0.01$, ***$p < 0.001$, and ****$p < 0.0001$.

All experiments have been reproduced in at least two independent experiments, unless otherwise specified in the figure legends. All immunoblots and images shown are the representative of these independent experiments.

**Reporting summary.** Further information on research design is available in the Nature Research Reporting Summary linked to this article.

## Data availability

Original data for IC$_{50}$ of chemotherapy drugs are available from GDSC (https://www.cancerrxgene.org/). mRNA expression data of SMARCA4/2 and ITPR3 are available from the Cancer Cell Line Encyclopedia (https://portals.broadinstitute.org/ccle) for cell lines and from UCSC Xena (https://xenabrowser.net/datapages/) for TCGA tumors of lung and ovarian cancer patients. Out of 13 SCCOHT patient tumors, RNA-seq data of ten cases were obtained from a previous study[87] and that of the other three cases can be found using the accession number EGAS00001005448. Source data for RNA-seq, microarray, ChIP-seq, and ATAC-seq can be found using the accession number GSE120297[49], GSE117735[60], GSE121755[48], GSE109010, and GSE109020[61]. All unique materials generated are readily available from the authors. All other data supporting the findings of this study are available from the corresponding author upon reasonable request. Source data are provided with this paper.

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

## Acknowledgements

The authors thank A. Leary for sharing RNA-seq data and R. Hass for sharing the SCCOHT-1 cells. The authors thank members of the Pelletier Laboratory for assistance with flow cytometry studies. This work was supported by Canadian Institutes of Health Research (CIHR) grants MOP-130540, PJT-156233, and PJT-438303 to S.H. and grant FDN-148390 to W.D.F. and a core grant from Medical Research Council (MRC) MC_UU_00015/7 to J.P. Author Y.X. was supported by CIHR Fellowship (MFE-171249), J.L.M. was supported by a MRC-funded graduate student fellowship, and S.H. was supported by a CRC Chair in Functional Genomics.

## Author contributions

Y.X., J.L.M., K.Y., Z.F., X.Z., B.M., L.W., G.M., A.M., and V.P. carried out experiments. Y.X. and J.L.M. performed statistical analyses. Y.X., F.J., and W.L. conducted bioinformatic analysis. A.Y., T.G., M.C., S.J., A.V.G., L.-M.P., J.S., and W.H.G. contributed samples and provided advice. P.O.F., S.C.-B., L.F., and M.-C.G. provided pathology expertise. Y.X., J.L.M., J.P., and S.H. wrote the manuscript with inputs from all authors. J.R., M.P., W.D.F., J.P., and S.H. supervised the experiments. J.P. and S.H. conceived and oversaw the study. All authors read and approved the final manuscript.

## Competing interests

The authors declare no competing interests.
