## [Peer Review File · Nature Communications]

REVIEWER COMMENTS

Reviewer #1 (Remarks to the Author):

This is a very nice and comprehensive study revealing a novel epigenetic mechanism by which IP3R3 expression is dysregulated in a subset of ovarian and lung cancers. The authors reveal key roles for SMARCA4/2 in chemotherapeutic responsiveness of cancer cells by driving IP3R3 expression. Loss of SMARCA4/2 results in loss of IP3R3 expression through reduced chromatin accessibility to the IP3R3 gene. Restoring SMARCA4/2 expression in cancer cell models restores IP3R3 expression and responsiveness to chemotherapy and Ca²⁺ driven apoptosis. Very importantly, pharmacological re-activation of SMARCA2 through a histone deacetylase inhibitor is sufficient to rescue IP3R3 expression and to enhance responsiveness to cisplatin. The work embraces several analyses including bio-informatic approaches, patient samples, RNAseq, unbiased CRISPR KO screen, functional calcium & cell death analysis and in vivo tumor growth. Overall, the combinatorial approaches provide very convincing evidence, yielding exciting and novel insights.

Comments:

1. It will be important to assess whether the effects of SMARCA4/2 are restricted to IP3R3 isoform alone or whether it also affects the expression of IP3R1 or IP3R2 channels. The Chip-Seq analyses implies not, but it would be good to have a validation through immunoblotting.
2. For several experiments, the cell death measurements are suboptimal. It would be important to provide cytometric analyses to have an idea of the extent of the cell death in the cell populations.
3. It seems that re-expression of SMARCA4 or SMARCA2 already provokes some cell death, cfr. cleaved caspase 3 and PARP in the absence of cisplatin (e.g. Fig1I in H1703 cells). Is this consistently observed? Is this related to IP3R3 expression?
4. Unfortunately, the authors have not measured Ca²⁺ responses towards cisplatin, whose actions seems to strongly depend on IP3R3 expression. It would be instrumental to assess whether cisplatin can indeed provoke pro-apoptotic Ca²⁺ rise that are dependent on IP3R3 expression.
5. i. Furthermore, several of the Ca²⁺ traces with histamine and particularly with thapsigargin (cfr supplemental figures) suggest changes in the decay of the Ca²⁺ signal. Could this be related to the concerted changes in ATP2B4, which encodes PMCA4. This is not further mentioned in the paper, but could actually impact some of the processes studied.
ii. Moreover, the quantification of the thapsigargin-releasable Ca²⁺ signals presented in Figure S8 is not complete. The authors should not only determine the peak value but also calculate the integrated signal (area under the curve). Moreover, re-expression of SMARCA4 has a clear impact on the decay phase of the thapsigargin-provoked Ca²⁺ signal. Thus, it is possible that concomitant upregulation of ATP2B4 (PMCA4) is responsible for enhanced Ca²⁺ clearance.
I would argue that deciphering the role of PMCA4 for chemotherapeutic responsiveness can be for another study, but at least, assessing the contribution of PMCA4 in the histamine, thapsigargin and cisplatin-provoked Ca²⁺ signals would be important.
6. The results with quisinostat are extremely exciting. However, they also raise a major concern as at the mechanistic level, they are less worked out. The authors should validate that the cisplatin/quisinostat-induced cell death is dependent on IP3R3 through siRNA experiments. It is also not clear whether this compound would be sufficient to augment Ca²⁺ signaling in response to agonists and/or cisplatin. This reviewer is particularly concerned, since the upregulation of IP3R3 is rather modest in this set of experiments, while the cell death effects are very striking. Moreover, quantification of the IP3R3-protein levels should be provided.
7. The paper would benefit from more quantitative analyses of the immunoblots, thereby presenting thereby presenting the results of quantifications from the independent experiments. This would be particularly important in the context of the cell death outcomes (in conjunction with other techniques).

Reviewer #2 (Remarks to the Author):

In this manuscript, Xue et al. discovered a new link between SMARCA4/2 loss associated chemo resistances to IP3R3 mediated mitochondrial Ca²⁺ influx. SMARCA4/2 play context-dependent roles in cancer development and are synthetic lethal in some cases but are co-deleted in some others including specific types of ovarian and lung cancers. This study nicely demonstrated a causal role of SMARCA4/2 loss in cisplatin resistance. It has been shown that cisplatin induced apoptosis depends on IP3R mediated ER to mitochondrial Ca²⁺ flux. This study further showed that SMARCA4/2 loss in certain cancer cell lines resulted in impaired ER-to-mitochondrial Ca²⁺ flux, possibly due to the reduced expression of ITPR3 gene. An HDAC inhibitor induced derepression of SMARCA2 in one cancer cell line could sensitize the cells and a xenograft model to cisplatin treatment. Thus, this study provides a mechanism underlying the chemo resistance of certain SMARCA4/2 deficient cancers, which may provide strategies for new cancer treatment. However, although IP3R3 could be one target gene regulating cisplatin resistance, it remains unclear whether other target genes or key pathways could be involved. A few questions need to be addressed to assess how general the mechanism revealed here is and whether alternative mechanisms exist.

1. The result showed that low expression of SMARCA4/2 is associated with less sensitivity to various chemotherapy treatments. This study focused on cisplatin. How about other chemo drugs? Can this IP3R3 mediated mitochondrial Ca²⁺ flux mechanism applied to other type of chemo resistance? Are there additional mechanisms?
2. SMARCA4/2 loss in ovarian and lung cancer cells affects cisplatin resistance. However, it is not clear whether this regulation of apoptosis could contribute to the tumor suppressor functions of SMARCA4/2 as shown in the cases of PTEN and BAP1. It may be helpful to discuss more.
3. The study suggests that ITPR3 is the main SMARCA4/2 target gene that mediate the function in Ca²⁺ flux and drug induced apoptosis. One important question is whether there are alternative explanations and whether there are other genes involved. The cross comparison of the RNA-seq data and ChIP-seq data narrowed down the candidate target genes. Since SMARCA4/2 ChIP signals sometimes are not distinct and could be missed, it may be worthwhile to just compare the ovarian and lung cancer RNA-seq data and not include the ChIP-seq comparison. Would that result in more candidate genes? How about the other three candidate SMARCA4/2 target genes (MATN2, EHD4 etc.)? Are they involved in chemo resistance? In addition, to exclude other possibilities, it would be important to examine other key regulators of mitochondrial Ca²⁺ influx such as other IP3R genes and Bcl family proteins. A gene expression comparison before and after cisplatin treatment in cells with different SMARCA4/2 levels would be helpful to understand the possible different functions of SMARCA4/2 in drug resistance.
4. The mitochondrial Ca²⁺ measurement was only performed upon histamine treatment. How about directly measure mitochondrial Ca²⁺ after cisplatin treatment?
5. In Figure 4 and supplemental figures, it seems that shR3 also reduced SMARCA4 level. Could the SMARCA4 dosage difference contribute to the phenotype reversal? The effects of different A4 dosages need to be examined. In addition, what is the control for shR3? Was the scrambled shRNA used as the control? If not, could this be an effect of shRNA expression?
6. It would be helpful to examine SMARCA2, IP3R3 levels and apoptotic markers in the xenograft tumors.

Reviewer #3 (Remarks to the Author):

In this manuscript, Xue et al., show that SMARCA4/2 deficiency, which is found in about 10% of patients with non-small cell lung cancer (NSCLC), and in virtually 100% of patients with small cell carcinoma of the ovary, hypercalcemic type (SCCOHT), is responsible for chemoresistance (to cisplatin) via inhibiting the expression levels of the IP3R3 Ca⁺⁺ pump in the ER. This results in deficient Ca⁺⁺ flux to the mitochondria with consequent inhibition of cisplatin-induced apoptosis in

cancer cells.

As outlined below, there are several issues and inconsistencies with this manuscript.

-In Figure 1 (C-G), the Authors knockdown SMARCA4/2 in the high-grade serous ovarian carcinoma derived OVCAR4 cell line to show that SMARCA4/2 loss results in resistance to cisplatin. This does not seem relevant since high-grade serous ovarian carcinoma is not characterized by loss of SMARCA4/2. These experiments should be conducted by knocking down SMARCA4/2 in SMARCA4/2 WT NSCLC cells.

-The same above is true for the experiments conducted in Figure 2 A-E. This needs to be done by knocking down SMARCA4/2 in SMARCA4/2 WT NSCLC cells.

-Authors should perform subcellular fractionation in SMARCA4/2 KD and OE cells and determine the differential expression of IP3R3 in the ER.

-The morphology of the IP3R3 #1 KD in supplemental Figure 11 is quite different as compared to control and KD#2. Can the author explain why?

-The in vivo experiments (Figure 5 K-M) needs to be done in PDX models from lung cancer patients with WT or mutated SMARCA4/2. These are available.

-Does Quisinostat+Cisplatin combination extend the overall survival in PDX models of SCCOHT? These also are available.

Reviewer #1 (Remarks to the Author):

This is a very nice and comprehensive study revealing a novel epigenetic mechanism by which IP3R3 expression is dysregulated in a subset of ovarian and lung cancers. The authors reveal key roles for SMARCA4/2 in chemotherapeutic responsiveness of cancer cells by driving IP3R3 expression. Loss of SMARCA4/2 results in loss of IP3R3 expression through reduced chromatin accessibility to the IP3R3 gene. Restoring SMARCA4/2 expression in cancer cell models restores IP3R3 expression and responsiveness to chemotherapy and Ca²⁺ driven apoptosis. Very importantly, pharmacological re-activation of SMARCA2 through a histone deacetylase inhibitor is sufficient to rescue IP3R3 expression and to enhance responsiveness to cisplatin. The work embraces several analyses including bio-informatic approaches, patient samples, RNAseq, unbiased CRISPR KO screen, functional calcium & cell death analysis and in vivo tumor growth. Overall, the combinatorial approaches are provide very convincing evidence, yielding exciting and novel insights.

We appreciate that the reviewer recognizes the novelty and contribution of our study and has provided us with helpful comments. We have carefully addressed all points raised in our revised manuscript. Please see our detailed response below.

Comments:

1. It will be important to assess whether the effects of SMARCA4/2 are restricted to IP3R3 isoform alone or whether it also affects the expression of IP3R1 or IP3R2 channels. The Chip-Seq analyses implies not, but it would be good to have a validation through immunoblotting.

Indeed, our previous analyses suggest that SMARCA4/2 specifically activates *ITPR3* but not *ITPR1* or *ITPR2* (please see also our response to point #3 of reviewer #2). We have now confirmed this with immunoblotting as shown in **Figure for Reviewer 1 (Fig. R1)**: restoration of SMARCA4/2 consistently upregulated IP3R3 but not IP3R1 or IP3R2 in SMARCA4/2-deficient ovarian and lung cancer cells (BIN-67, SCCOHT-1 and H1703). Please note that IP3R2 has low expression in cell lines tested and we validated IP3R1/2 antibodies using shRNAs as shown in **Fig. R2**.

In addition, we performed functional experiments and found that shRNA-mediated suppression of IP3R1 or IP3R2 in SMARCA4/2-proficient OVCAR4 cells did not significantly impact cisplatin-induced apoptosis, as indicated by expression of cleaved PARP and cleaved caspase-3, compared to the dramatic effect of shRNA-mediated suppression of IP3R3 (**Fig. R2**). These results support that IP3R3 is the dominant regulator of Ca²⁺-mediated apoptotic response in our model systems.

2. For several experiments, the cell death measurements are suboptimal. It would be important to provide cytometric analyses to have an idea of the extent of the cell death in the cell populations.

We agree with the reviewer and have now included flow cytometry data for the relevant experiments to provide a better quantification of cell death, which are consistent with our previous results. Please see **new Fig. 1F, 1I, 4H, 4K, 6C**.

3. It seems that re-expression of SMARCA4 or SMARCA2 already provokes some cell death, cfr. cleaved caspase 3 and PARP in the absence of cisplatin (e.g. Fig1I in H1703 cells). Is this consistently observed? Is this related to IP3R3 expression?

We thank the reviewer for pointing this out. Indeed, we consistently observed that re-expression of SMARCA4/2 induces some levels of cell death in the absence of cisplatin.

Related to original Fig1I (**Now Fig. 1H**), we have performed quantification of western blots (**new Fig. S16A**) and flow cytometry analysis (**new Fig. 1I**) from independent experiments, which are all consistent with this induction of cell death in the absence of cisplatin upon SMARCA4/2 restoration. Please see below for the compilation of these data.

Fig. 1H

Fig. 1I

Fig. S16A

As well, as shown in our **original Fig. 4J**, SMARCA4 restoration in H1703 led to elevation of cleaved PARP and cleaved caspase 3 both before (and after) cisplatin treatment, which was associated with the concomitant increase of IP3R3 expression; knockdown of IP3R3 markedly suppressed the induction of these apoptosis markers in these SMARCA4-expressing cells. We have performed quantification of western blots (**new Fig. S16B**) and flow cytometry analysis (**new Fig. 4K**) from independent experiments. Please see below for the compilation of these data.

Fig. 4J

Fig. 4K

Fig. S16B

Furthermore, we also obtained similar results *in vivo*. As shown below and in **new Fig. 5I, J, K**, forced expression of SMARCA4 using a Tet-on inducible system in H1703 xenografts led to significant growth suppression, associated with elevation of IP3R3 and cleaved caspase 3 expression, as indicated by immunohistochemistry (IHC). Together, these results support the notion that IP3R3 dysregulation may contribute to the tumorigenesis driven by SMARCA4 loss through inhibiting apoptosis.

Fig. 5

4. Unfortunately, the authors have not measured Ca²⁺ responses towards cisplatin, whose actions seems to strongly depend on IP3R3 expression. It would be instrumental to assess whether cisplatin can indeed provoke pro-apoptotic Ca²⁺ rise that are dependent on IP3R3 expression.

We agree with the reviewer and have conducted two new sets of experiments to address this comment. First, we now showed that quisinostat treatment of H1703 cells led to an increase of cytosolic and mitochondrial Ca²⁺ contents upon histamine stimulation (**New Fig. S14**; see also below). Importantly, we showed that this rescue in Ca²⁺ signalling was dependent on SMARCA4 and IP3R3, since silencing of IP3R3 decreased both cytosolic and mitochondrial Ca²⁺ levels in quisinostat treated and SMARCA4-restored H1703 cells. These results confirm that the Ca²⁺ response induced by quisinostat treatment is SMARCA4 and IP3R3 dependent.

Fig. S14 A

B

C

In addition, we also have performed confocal live cell imaging of H1703 cells expressing the mitochondrial Ca^{2+} probe CEPIA-2mt and labelled with Mitotracker and showed that treatment with quisinostat and cisplatin induced a significant increase of the basal mitochondrial Ca^{2+} pool (New Fig. 6G, H; see also below). Together these data confirmed our cell death assay showing the dependence of apoptosis for A2/A4 and IP3R3 upon cisplatin and quisinostat treatments (Fig. 6A-F).

Fig. 6

5. i. Furthermore, several of the Ca^{2+} traces with histamine and particularly with thapsigargin (cfr supplemental figures) suggest changes in the decay of the Ca^{2+} signal. Could this be related to the concerted changes in ATP2B4, which encodes PMCA4. This is not further mentioned in the paper, but could actually impact some of the processes studied.

ii. Moreover, the quantification of the thapsigargin-releasable Ca^{2+} signals presented in Figure S8 is not complete. The authors should not only determine the peak value but also calculate the integrated signal (area under the curve). Moreover, re-expression of SMARCA4 has a clear impact on the decay phase of the thapsigargin-provoked Ca^{2+} signal. Thus, it is possible that concomitant upregulation of ATP2B4 (PMCA4) is responsible for enhanced Ca^{2+} clearance.

I would argue that deciphering the role of PMCA4 for chemotherapeutic responsiveness can be for another study, but at least, assessing the contribution of PMCA4 in the histamine, thapsigargin and cisplatin-provoked Ca^{2+} signals would be important.

As requested, we have determined the integrated signal including area under the curve (AUC) for the original Fig. S6 and S8 (now **new Fig. S7** and **S9**). These new analyses confirmed our previous results that SMARCA4 knockout in OVCAR4 cells exhibited an increase of ER- Ca^{2+} store measured by thapsigargin treatment (**new Fig. S7**). However, these cells being resistant to apoptosis induction, it is unlikely that this slight increase of ER- Ca^{2+} is involved in the resistance to cell death observed in this condition. Conversely, we observed a small decrease of ER- Ca^{2+} content measured by thapsigargin treatment in SCCOHT-1 and H1703 cells restored with SMARCA4 expression (**new Fig. S9**). We have shown that A4 restoration in these cells were required to induce cell death by IP3R3-mediated mitochondrial Ca^{2+} , indicating that the quantity of Ca^{2+} in the ER, even if slightly lower, is sufficient to induce cell death.

In this revision, we also further validated our findings using additional NSCLC cell line H1437 (**new Fig. 2L-N, S2D**). Please see also our response to *points #1, 2 of Reviewer #3*.

We agree that PMCA4 regulation by SMARCA4 likely contributed to the decade of Ca^{2+} signal in SMARCA4-restored cells. We have confirmed the regulation of PMCA4 by SMARCA4/2 by immunoblots (see below **Fig. R3**).

Given that the role of PMCA4 in extruding Ca^{2+} across the plasma membrane, reduced PMCA4 expression in SMARCA4-deficient cells is expected to result in increase of cytosolic Ca^{2+} . This would not contribute to the significant reduced Ca^{2+} flux (caused by decreased IP3R3 expression) into the mitochondria we observed upon agonist stimulation.

Nevertheless, we assessed the role of PMCA4 in mediating chemotherapeutic responsiveness and knocked down PMCA4 using two independent shRNAs in SMARCA4/2-proficient OVCAR4 (ovarian) and H1437 (NSCLC) cells. Suppression of PMCA4 did not results in significant changes in expression of cleaved PARP and cleaved Caspase 3 in the absence or the presence of cisplatin in both cell lines (see below **Fig. R4**). These results support the dominant role of IP3R3 in mediating chemotherapeutic responsiveness in our study.

However, we agree with the reviewer that dysregulation of PMCA4 (and other SMARCA4/2 targets) may contribute to altered Ca^{2+} signalling in SMARCA4-deficient cancer which warrants further future study. We have also modified our discussion to reflect this point.

6. The results with quisinostat are extremely exciting. However, they also raise a major concern as at the mechanistic level, they are less worked out. The authors should validate that the cisplatin/quisinostat-induced cell death is dependent on IP3R3 through siRNA experiments. It is also not clear whether this compound would be sufficient to augment Ca^{2+} signaling in response to agonists and/or cisplatin. This reviewer is particularly concerned, since the upregulation of IP3R3 is rather modest in this set of experiments, while the cell death effects are very striking. Moreover, quantification of the IP3R3-protein levels should be provided.

We agree with the reviewer that this is an important control experiment and have conducted the study. As shown below and in **new Fig. 6F**, siRNA-mediated suppression of IP3R3 indeed blunted the induction of cleaved PARP and cleaved caspase 3 expression upon cisplatin/quisinostat treatment. These data support that IP3R3 is required for the apoptotic induction by this drug combination. In addition, we have quantified the IP3R3 protein expression in the relevant blots (see **new Figure S16**).

Furthermore, as already presented in our response to *point#4 of Reviewer #1* above, we have now performed two new sets of calcium experiments to show that quisinostat treatment is sufficient to increase ER-induced mitochondrial Ca^{2+} levels in a SMARCA4 and IP3R3-dependent manner, and that its combination with cisplatin increased significantly basal mitochondrial Ca^{2+} pools (Please see **new Fig. S14** and **new Fig. 6G, H**).

Moreover, we also included new data showing that CRISPR/Cas9-mediated knockout of SMARCA2 in H1703 cells suppressed the induction of IP3R3, cleaved PARP and cleaved caspase 3 expression upon cisplatin/quisinostat treatment (see below **new Figure 6E**).

Together, these new control experiments further support the notion that cisplatin/quisinostat-induced cell death is dependent on the SMARCA2-IP3R3 axis.

7. The paper would benefit from more quantitative analyses of the immunoblots, thereby presenting thereby presenting the results of quantifications from the independent experiments. This would be particularly important in the context of the cell death outcomes (in conjunction with other techniques).

We agree with the reviewer and we have now quantified the immunoblots from independent experiments as suggested through all the manuscript (see **new Figure S16**).

Reviewer #2 (Remarks to the Author):

In this manuscript, Xue et al. discovered a new link between SMARCA4/2 loss associated chemo resistances to IP3R3 mediated mitochondrial Ca²⁺ influx. SMARCA4/2 play context-dependent roles in cancer development and are synthetic lethal in some cases but are co-deleted in some others including specific types of ovarian and lung cancers. This study nicely demonstrated a causal role of SMARCA4/2 loss in cisplatin resistance. It has been shown that cisplatin induced apoptosis depends on IP3R mediated ER to mitochondrial Ca²⁺ flux. This study further showed that SMARCA4/2 loss in certain cancer cell lines resulted in impaired ER-to-mitochondrial Ca²⁺ flux, possibly due to the reduced expression of ITPR3 gene. An HDAC inhibitor induced derepression of SMARCA2 in one cancer cell line could sensitize the cells and a xenograft model to cisplatin treatment. Thus, this study provides a mechanism underlying the chemo resistance of certain SMARCA4/2 deficient cancers, which may provide strategies for new cancer treatment. However, although IP3R3 could be one target gene regulating cisplatin resistance, it remains unclear whether other target genes or key pathways could be involved. A few questions need to be addressed to access how general the mechanism revealed here is and whether alternative mechanisms exist.

We thank the reviewer for recognizing the novelty and potential clinical implication of our study and for providing the constructive comments. We have carefully addressed all points raised in our revised manuscript. Please also see our detailed response below.

1. The result showed that low expression of SMARCA4/2 is associated with less sensitivity to various chemotherapy treatments. This study focused on cisplatin. How about other chemo drugs? Can this IP3R3 mediated mitochondrial Ca²⁺ flux mechanism applied to other type of chemo resistance? Are there additional mechanisms?

We have conducted key experiments using additional chemotherapeutics commonly used in the clinic, including cyclophosphamide, topotecan and paclitaxel. As shown below (**new Fig. S4A, B**), SMARCA4 knockout in HEC116 cells also inhibited the induction of cleaved PARP and cleaved caspase 3 by these additional agents and increased cell viability upon these treatments; conversely, restoration of SMARCA4/2 sensitized H1703 cells to apoptosis induction and growth inhibition by these chemo drugs (**new Fig. S4C, D**). These new data are consistent with our original findings showing that *SMARCA4/2* expression correlates with resistance to different chemotherapies (**Original Fig. S2**) and the paclitaxel data obtained in OVCAR4 cells (**Original Fig. S3a**).

While we appreciate the reviewer's comments to elucidate if these different chemotherapies are, in addition to SMARCA4/2 expression (described above), also IP3R3-mediated Ca²⁺ dependent. However, to analyse IP3R3-dependence in cell death and Ca²⁺ assays represented an extensive new set of experiments that was not able to be performed due to time restriction during the revision process. While our current study indicates that IP3R3 dysregulation underlies apoptotic resistance in SMARCA4/2 deficient cancer cells, we agree with the reviewer that our data do not rule out additional mechanisms which require future investigations. We have addressed this point in the discussion.

Figure S4

2. SMARCA4/2 loss in ovarian and lung cancer cells affects cisplatin resistance. However, it is not clear whether this regulation of apoptosis could contribute to the tumor suppressor functions of SMARCA4/2 as shown in the cases of PTEN and BAP1. It may be helpful to discuss more.

We fully agree with the reviewer on this point and have revised our discussion as suggested. Further supporting this point, we have now provided new *in vivo* data showing that re-expression of SMARCA4 suppressed tumor growth of H1703 xenografts with increased IP3R3 and cleaved caspase 3 expression (New Fig. 5I, J, K). Please see also our response to *point #3 of Reviewer #1*.

3. The study suggests that ITPR3 is the main SMARCA4/2 target gene that mediate the function in Ca²⁺ flux and drug induced apoptosis. One important question is whether there are alternative explanations and whether there are other genes involved. The cross comparison of the RNA-seq data and ChIP-seq data narrowed down the candidate target genes. Since SMARCA4/2 ChIP signals sometimes are not distinct and could be missed, it may be worthwhile to just compare the ovarian and lung cancer RNA-seq data and not include the ChIP-seq comparison. Would that result in more candidate genes? How about the other three candidate SMARCA4/2 target genes (MATN2, EHD4 etc.)? Are they involved in chemo resistance? In addition, to exclude other possibilities, it would be important to examine other key regulators of mitochondrial Ca²⁺ influx such as other IP3R genes and Bcl family proteins. A gene expression comparison before and after cisplatin treatment in cells with different SMARCA4/2 levels would be helpful to understand the possible different functions of SMARCA4/2 in drug resistance.

We agree with the reviewer that our data do not rule out other potential mechanisms in addition to ITPR3 regulation. We used ChIP-seq comparison to prioritize the direct targets of SMARCA4/2 leading to the identification of ITPR3, which formed the focus of our current study. While we believe that other potential candidates require future investigations which is beyond the scope of this study, we have performed additional experiments and analysis as suggested.

We examined the 198 genes related calcium signaling that were regulated by SMARCA4 in SCCOHT cells, without using the ChIPseq data for comparison. This yielded 13 genes that were also significantly regulated by SMARAC4 in H1299 NSCLC cells, 7 of which was regulated in the same direction (See below Fig. R5). These candidates may serve suggestions for future studies and we have modified our discussion to reflect this.

Fig. R5

We also examined the regulation of other IP3Rs and BCL-2 family genes in the RNAseq data without including the ChIPseq data set. As shown below in the **Fig. R6** below, we did not find additional genes that were consistently regulated by SMARCA4 in both ovarian and lung cancer models, besides *ITPR3*. In addition, we performed functional studies for IP3R1 and IP3R2 and found that their suppression also did not significantly impact apoptosis (please also see our response to *point #1 of Reviewer #1*, **Fig. R1, R2**).

Regarding other SMARCA4/2 target genes candidates from our original analyses, we have accessed the potential role of ATP2B4 (PMCA4) since it has a direct role in extruding Ca^{2+} across the plasma membrane. As described in our response to *point #5 of Reviewer #1*, we have validated the regulation and also performed RNAi-mediated knockdown in SMARCA4 proficient OVCAR4 (ovarian) and H1437 (NSCLC) cells. Suppression of PMCA4 did not impact apoptosis in the absence or the presence of cisplatin in both cell lines (see **Fig. R3, R4**). These results support the dominant role of IP3R3 in mediating chemotherapeutic responsiveness in our current study.

Furthermore, we also analysed a publicly available gene expression data set in NSCLC cell lines before and after cisplatin treatment (GSE116439). Among IP3Rs and BCL-2 family genes, the only one that was differential regulated depending on the SMARCA4 status was *PMAIP1* (*NOXA*), which belongs to the pro-apoptotic BH3-only subfamily (See below **Fig. R7**). This dysregulation of *NOXA* may also contribute to reduced apoptosis in SMARCA4 loss, which requires further investigation but beyond the scope of our current study focusing on the regulation between SMARCA4/2 and IP3R3. However, we fully agree with the reviewer that it is important to acknowledge other potential mechanisms and have revised our discussion to reflect this.

Fig. R7

4. The mitochondrial Ca²⁺ measurement was only performed upon histamine treatment. How about directly measure mitochondrial Ca²⁺ after cisplatin treatment?

We thank the reviewer for this comment, and experiments have now been performed to analyse basal mitochondrial Ca²⁺ pool after quisinostat, cisplatin and combination. Please see our answer to *point#4 of Reviewer#1*.

5. In Figure 4 and supplemental figures, it seems that shR3 also reduced SMARCA4 level. Could the SMARCA4 dosage difference contribute to the phenotype reversal? The effects of different A4 dosages need to be examined. In addition, what is the control for shR3? Was the scrambled shRNA used as the control? If not, could this be an effect of shRNA expression?

We thank the reviewer for raising these valid points and have performed the suggested experiments to address these concerns. Using an inducible SMARCA4 expression construct, we tested the different dosages of SMARCA4 re-expressing in H1703 cells with different doses of doxycycline. As shown below in **Fig. R8**, the minimum level of SMARCA4 re-expression is sufficient to induce maximum IP3R3 expression. Similar regulation was also observed for other key SMARCA4 target such as cyclin D1 in our previously published study (PMID 30718512, Fig. S10A, E). Consistently, differential expression levels of SMARCA4 expression had similar effect on cleaved PARP and Caspase 3 (**Fig. R8B**).

Regarding the control for sh*ITPR3*, we apologize for not describe clearly and have clarified this in Methods of the revised manuscript. The pLKO empty vector was used in these studies at low MOI same as with the shRNAs. To rule out the potential effects of the shRNA expression, we have performed experiment in OVCAR4 and H1703 cells and compared the effect of empty vector, a scrambled shRNA and a shRNA targeting luciferase. As shown below in **Fig. R9**, there was no difference observed in IP3R3 expression levels in both cell lines expressing these different controls.

6. It would be helpful to examine SMARCA2, IP3R3 levels and apoptotic markers in the xenograft tumors.

We agree with the reviewer and have conducted these IHC experiments as suggested. As shown below and in **new Fig. 6I, J, K**, the results obtained are consistent with the tumor growth suppression observed.

Fig. 6

Reviewer #3 (Remarks to the Author):

In this manuscript, Xue et al., show that SMARCA4/2 deficiency, which is found in about 10% of patients with non-small cell lung cancer (NSCLC), and in virtually 100% of patients with small cell carcinoma of the ovary, hypercalcemic type (SCCOHT), is responsible for chemoresistance (to cisplatin) via inhibiting the expression levels of the IP3R3 Ca⁺⁺ pump in the ER. This results in deficient Ca⁺⁺ flux to the mitochondria with consequent inhibition of cisplatin-induced apoptosis in cancer cells.

We appreciate that the reviewer for carefully reviewing our manuscript and for providing us with helpful suggestions. We have carefully addressed the points raised. Please see our detailed response below.

As outlined below, there are several issues and inconsistencies with this manuscript.

-In Figure 1 (C-G), the Authors knockdown SMARCA4/2 in the high-grade serous ovarian carcinoma derived OVCAR4 cell line to show that SMARCA4/2 loss results in resistance to cisplatin. This does not seem relevant since high-grade serous ovarian carcinoma is not characterized by loss of SMARCA4/2. These experiments should be conducted by knocking down SMARCA4/2 in SMARCA4/2 WT NSCLC cells.

We agree that HGSC is not characterized with SMARCA4/2 loss. The isogenic cell pair created using OVCAR4 was to demonstrate SMARCA4/2 loss can result in chemoresistance in the context of ovarian cancer. Similar results were also obtained in HEC116 endometrial adenocarcinoma cells (**original Fig. S3**).

As suggested, we have generated SMARCA4 knockout in SMARCA4/2-proficient H1437 NSCLC cells and also further knocked down SMARCA2 using shRNAs. As shown in **New Fig. S3G, H**, *SMARCA4* knockout cells were more resistant to cisplatin-induced elevation of cleaved PARP and cleaved caspase 3; knockdown of *SMARCA2* in these SMARCA4 knockout cells led to increased resistance to these apoptotic responses induced by cisplatin. *SMARCA4* knockout in H1437 cells also led to decreased IP3R3 expression (**new Fig. S8D**).

Fig. S3

Fig. S8

-The same above is true for the experiments conducted in Figure 2 A-E. This needs to be done by knocking down SMARCA4/2 in SMARCA4/2 WT NSCLC cells.

We have performed the experiments analysing the Ca^{2+} flux in SMARCA4/2-proficient H1437 NSCLC cells SMARCA4 knockout. As shown in **the new Fig. 2 L-N, S7D** and below, we obtained the similar results as observed in OVCAR4 cells \pm SMARCA4 knockout (**original Fig. 2I-K, S7C**). Indeed, we confirmed that SMARCA4 knockout led to a reduced ER-induced cytosolic Ca^{2+} pool, accompanied by a significant reduction of the mitochondrial Ca^{2+} levels, upon histamine stimulation.

Fig. 2

Fig. S7

-Authors should perform subcellular fractionation in SMARCA4/2 KD and OE cells and determine the differential expression of IP3R3 in the ER.

To perform ER fractionation, we followed the manufacturer's protocol using a frequently cited ER isolation kit (Sigma Cat# ER0100) which requires >200 million cells for each condition. This enormous number of cells is not feasible for SMARCA4/2 overexpression experiments considering replicates with multiple conditions – SCCOHT cells cannot tolerate SMARCA4/2 restoration; forced SMARCA4/2 expression in NSCLC cells such as H1703 also suppress their growth with passage expansion. However, we were able to perform the reciprocal experiments in HEC116 cell pairs with or without SMARCA4 knockout (300 million cells for each condition). As shown below in **Fig. R10**, both basal (parental) and elevated (SMARCA4 knockout) IP3R3 expression were detected in ER fraction but not in the cytosolic control. This is consistent with the literature of well-established ER localization of IP3R3.

-The morphology of the IP3R3 #1 KD in supplemental Figure 11 is quite different as compared to control and KD#2. Can the author explain why?

To validate the IP3R3 antibody for IHC staining, we used cell pellets which can display different levels of cell compaction following sample processing and paraffin embedding. To illustrate this experimental variation underlying this difference in morphology in the original Fig. S11B (**now Fig. S12B**, also below), we have taken additional images from different portions of the same cell pellets showing below in **Fig. R11**.

Fig. S12B

Fig. R11

-The *in vivo* experiments (Figure 5 K-M) needs to be done in PDX models from lung cancer patients with WT or mutated SMARCA4/2. These are available.

-Does Quisinostat+Cisplatin combination extend the overall survival in PDX models of SCCOHT? These also are available.

We have searched in the literature and commercial sources but failed to identify a lung PDX model that is SMARCA4 mutated also with confirmed SMARCA2 epigenetic silencing - this is required for quisinostat-mediated SMARCA2/ITPR3 reactivation.

Commercial PDX sources including Charles River (~\$35,000 USD for 32 mice (4 arms)/model for contract service) and The Jackson Laboratory (~\$15,000 USD for 32 mice/model purchasable) do offer SMARCA4-mutated NSCLC PDXs, but their SMARCA2 epigenic status is not available. While we could identify those that express low levels of SMARCA2 (RNAseq/IHC), this may be due to other mechanisms that do not allow quisinostat-mediated of SMARCA2 reactivation. We would have to treat each available SMARCA4-mutated PDX with quisinostat and examine SMARCA2 expression. This is not feasible considering the high cost and unpredictable growth rates of different PDXs (4-6 months for cryorecovery followed by 1-2 months of passaging for experimental cohorts).

We were unable to access SCCOHT PDXs, and as far as we are aware, none are commercially available. However, for this revision, we have provided additional *in vivo* experiment using H1703 xenograft model further supporting the novel link between SMARCA4 loss, IP3R3 and apoptosis (please see our response to **point #3 of Reviewer #1** and also **new Fig. 5I, J, K**). Except for the PDX request due to the restrictions described above, we have completed all other experiments suggested by all reviewers. We believe that our revised manuscript has improved significantly and strengthened the mechanistic link between reduced IP3R3-mediated Ca²⁺ flux and chemotherapy resistance induced by SMARCA4/2 loss in both ovarian and lung cancers.

REVIEWERS' COMMENTS

Reviewer #1 (Remarks to the Author):

The authors have provided an adequate response to my comments with substantial novel experiments that take away any reservations about the work and further underpin the conclusions. I am very excited about these findings and novel insights brought forward. The work is of very high quality and deserves to be published in Nature Communications journal.

Reviewer #2 (Remarks to the Author):

The authors have addressed most if not all the issues raised by me.

Reviewer #3 (Remarks to the Author):

The Authors have addressed satisfactorily my set of comments therefore I recommend this manuscript for publication.

REVIEWERS' COMMENTS

Reviewer #1 (Remarks to the Author):

The authors have provided an adequate response to my comments with substantial novel experiments that take away any reservations about the work and further underpin the conclusions. I am very excited about these findings and novel insights brought forward. The work is of very high quality and deserves to be published in Nature Communications journal.

Reviewer #2 (Remarks to the Author):

The authors have addressed most if not all the issues raised by me.

Reviewer #3 (Remarks to the Author):

The Authors have addressed satisfactorily my set of comments therefore I recommend this manuscript for publication.

We are pleased that all reviewers are satisfied by our revised manuscript. We would like to thank them for carefully and fairly evaluating our studies and for providing the constructive comments, which have helped us improving our manuscript.